



# The penultimate deglaciation: protocol for PMIP4 transient numerical simulations between 140 and 127 ka

Laurie Menviel[1,2,*], Emilie Capron[3,4,*], Aline Govin[5], Andrea Dutton[6], Lev Tarasov[7], Ayako Abe-Ouchi[8], Russell Drysdale[9,10], Philip Gibbard[11], Lauren Gregoire[12], Feng He[13], Ruza Ivanovic[12], Masa Kageyama[5], Kenji Kawamura[14,15,16], Amaelle Landais[5], Bette L. Otto-Bliesner[17], Ikumi Oyabu[14], Polychronis Tzedakis[18], Eric Wolff[19], and Xu Zhang[20,21]

[1]Climate Change Research Center, PANGEA, ARC Centre of Excellence in Climate System Science, the University of New South Wales, Sydney, Australia
[2]Department of Earth and Planetary Sciences, Macquarie University, Sydney, NSW 2109, Australia
[3]Centre for Ice and Climate, Niels Bohr Institute, University of Copenhagen, Juliane Maries Vej 30, Copenhagen, DK-2900, Denmark
[4]British Antarctic Survey, High Cross, Madingley Road, Cambridge, CB3 0ET, UK
[5]Laboratoire des Sciences du Climat et de l'Environnement (LSCE), Institut Pierre Simon Laplace (IPSL), CEA-CNRS-UVSQ, Université Paris-Saclay, Gif-Sur-Yvette, 91190, France
[6]Department of Geological Sciences, University of Florida, PO Box 112120, Gainesville, FL 32611, USA
[7]Department of Physics and Physical Oceanography, Memorial University of Newfoundland, St John's, Canada
[8]Atmosphere and Ocean Research Institute, The University of Tokyo, Tokyo, Japan
[9]School of Geography, The University of Melbourne, Melbourne, Australia
[10]Laboratoire EDYTEM UMR CNRS 5204, Université Savoie Mont Blanc, 73376 Le Bourget du Lac, France
[11]Scott Polar Research Institute, University of Cambridge, Cambridge, CB2 1ER, UK
[12]School of Earth and Environment, University of Leeds, Leeds, LS2 9JT, UK
[13]Center for Climatic Research, Nelson Institute for Environmental Studies, University of Wisconsin-Madison, Madison, WI 53706, USA
[14]National Institute of Polar Research, Research Organizations of Information and Systems,10-3 Midori-cho, Tachikawa, Tokyo 190-8518, Japan
[15]Department of Polar Science, Graduate University for Advanced Studies (SOKENDAI), 10-3 Midori-cho, Tachikawa, Tokyo 190-8518, Japan
[16]Institute of Biogeosciences, Japan Agency for Marine-Earth Science and Technology, 2-15 Natsushima-cho, Yokosuka 237-0061, Japan
[17]Climate and Global Dynamics Laboratory, National Center for Atmospheric Research (NCAR), Boulder, CO 80305, USA
[18]Environmental Change Research Centre, Department of Geography, University College London, London, UK
[19]Department of Earth Sciences, University of Cambridge, Cambridge, CB2 3EQ, UK
[20]Alfred Wegener Institute, Helmholtz Centre for Polar and Marine Research, D-27570 Bremerhaven, Germany
[21]Key Laboratory of Western China's Environmental Systems (Ministry of Education), College of Earth and Environmental Sciences, Lanzhou University, Lanzhou, 730000, China
[*]Both authors contributed equally to this work

Correspondence: L. Menviel (l.menviel@unsw.edu.au) and E. Capron (ecap@bas.ac.uk)

**Abstract.** The penultimate deglaciation (∼138-128 thousand years before present, hereafter ka) is the transition from the penultimate glacial maximum to the Last Interglacial (LIG, ∼129-116 ka). The LIG stands out as one of the warmest interglacials of the last 800 ka, with high-latitude temperature warmer than today and global sea level likely higher by at least 6 meters. The LIG therefore receives ever-growing attention, in particular to identify mechanisms and feedbacks responsible for such





regional warmth that is comparable to that expected before 2100. Considering the transient nature of the Earth system, the LIG climate and ice-sheets evolution were certainly influenced by the changes occurring during the penultimate deglaciation. It is thus important to investigate the climate and environmental response to the large changes in boundary conditions (i.e. orbital configuration, atmospheric greenhouse gas concentrations, ice sheet geometry) occurring during this time interval.

A deglaciation working group has recently been set up as part of the Paleoclimate Modelling Intercomparison Project (PMIP) phase 4, with a protocol to perform transient simulations of the last deglaciation (19-11 ka). Similar to the last deglaciation, the disintegration of continental ice-sheets during the penultimate deglaciation led to significant changes in the oceanic circulation during Heinrich Stadial 11 (∼136-129 ka). However, the two deglaciations bear significant differences in magnitude and temporal evolution of climate and environmental changes.

Here, as part of the PAGES-PMIP working group on Quaternary Interglacials, we propose a protocol to perform transient simulations of the penultimate deglaciation to complement the PMIP4 effort. This design includes time-varying changes in orbital forcing, greenhouse gas concentrations, continental ice-sheets as well as freshwater input from the disintegration of continental ice-sheets. This experiment is designed to assess the coupled response of the climate system to all forcings. Additional sensitivity experiments are proposed to evaluate the response to each forcing. Finally, a selection of paleo records

representing different parts of the climate system is presented, providing an appropriate benchmark for upcoming model-data comparisons across the penultimate deglaciation.

# 1 Introduction

Over the last 450 thousand years (kyr), Earth's climate has been dominated by glacial-interglacial cycles with a recurrence period of about 100 kyr. These asymmetrical cycles are characterised by long glacial periods, associated with a gradual cooling,

a slow decrease in atmospheric greenhouse gas (GHG) concentrations and a progressive growth of large continental ice sheets in the Northern Hemisphere (NH), leading to a 60 to 120 m global sea-level decrease (Lisiecki and Raymo, 2005; Grant et al., 2014; Rohling et al., 2017). These long glacial periods were followed by relatively rapid multi-millennial-scale warmings into an interglacial state. These transitions from a glacial to an interglacial climate are also referred to as deglaciations or terminations. While the term "termination" was originally defined as the midpoint of the multi-millennial-scale glacial-interglacial

transition in marine foraminifera stable oxygen isotopes (Broecker and van Donk, 1970), we will use this term here to refer to the full glacial-interglacial transition.

Glacial terminations are paced by an external forcing, i.e. variations in the seasonal and latitudinal distribution of incoming solar radiation (insolation) driven by changes in Earth's orbit (Berger, 1978). However, changes in insolation alone are not sufficient to explain the amplitude of these major warmings and require amplification mechanisms. These amplification

mechanisms are related to (i) the large increase in atmospheric GHG concentrations, e.g. atmospheric $CO_2$ increases by 60 to 100 ppm (Lüthi et al., 2008) (Fig. 1), and (ii) the disintegration of NH ice-sheets and their associated change in albedo (Abe-Ouchi et al., 2013).



A pervasive characteristic of glacial terminations of the past 450 kyr is the occurrence of millennial-scale climate events (e.g. Cheng et al., 2009; Barker et al., 2011; Vázquez Riveiros et al., 2013; Past Interglacials Working Group of PAGES, 2016; Rodrigues et al., 2017). In the North Atlantic, these events, also referred to as stadials, are characterised by a substantial weakening of North Atlantic Deep Water (NADW) formation (e.g. McManus et al., 2004; Vázquez Riveiros et al., 2013;

Böhm et al., 2015; Ng et al., 2018), possibly due to meltwater discharge into the North Atlantic (Ivanovic et al., 2018). There is a link between these events and enhanced iceberg calving, supported by the presence of Ice Rafted Debris (IRD) in North Atlantic marine sediment cores, and the stadials that contain substantial IRD layers (identified as Heinrich Events) are known as Heinrich Stadials (e.g. Heinrich, 1988; Bond et al., 1992; McManus et al., 1999; van Kreveld et al., 2000; Hodell et al., 2017).

A weakening of the Atlantic meridional heat transport during these stadials maintains cold conditions at high northern lat-

itudes in the Atlantic sector (Stouffer et al., 2007; Swingedouw et al., 2009; Kageyama et al., 2010, 2013) while contributing to a gradual warming at high southern latitudes (Blunier and Brook, 2001; Stocker and Johnsen, 2003; EPICA and community members, 2006), thus leading to a bipolar-seesaw pattern of climate changes. The strengthening of NADW formation at the end of stadials induces a relatively abrupt temperature increase in the North Atlantic and surrounding regions, and a sharp increase in atmospheric $CH_4$ and Asian monsoon strength (Loulergue et al., 2008; Cheng et al., 2009; Buizert et al., 2014).

While a significant atmospheric $CO_2$ increase is also observed during the NADW strengthening at the end of Heinrich stadials during deglaciations (Marcott et al., 2014), the major phase of atmospheric $CO_2$ increase coincides with a Southern Ocean (e.g. Barker et al., 2009; Uemura et al., 2018) and Antarctic warming (Cheng et al., 2009; Masson-Delmotte et al., 2011; Landais et al., 2013; Marcott et al., 2014) (Fig. 1). The sequence of events leading to the deglacial atmospheric $CO_2$ increase is still poorly constrained. Still, it most likely resulted from a combination of processes (e.g. Kohfeld and Ridgwell,

2009), including changes in solubility, global alkalinity content (e.g. Sigman et al., 2010), iron fertilization (e.g. Martin, 1990; Bopp et al., 2003; Martinez-Garcia et al., 2014), Antarctic sea ice cover (Stephens and Keeling, 2000), and changes in ocean circulation (e.g. Toggweiler et al., 2006; Anderson et al., 2009; Skinner et al., 2010; Toggweiler and Lea, 2010). Changes in ocean circulation, and particularly variations in the formation rates of the main deep and bottom water masses, i.e. NADW and Antarctic Bottom Water, can significantly impact atmospheric $CO_2$ by modifying the vertical gradient in oceanic dissolved

inorganic carbon (e.g. Menviel et al., 2014, 2017, 2018).

While they share similarities, past deglaciations also bear significant differences in amplitude and durations. They initiate under a range of glacial ice-sheet states and progress under a variety of orbital-forcing scenarios (Past Interglacials Working Group of PAGES, 2016; Tzedakis et al., 2017) (Fig. 1). Although there are still many open questions, the sequence of events occurring during the last deglaciation (referred here as TI for Termination I), which represents

the transition from the Last Glacial Maximum (LGM, Marine Isotope Stage 2, hereafter MIS2, 26-19 ka) to our current interglacial, is starting to emerge (Cheng et al., 2009; Denton et al., 2010; Shakun et al., 2012). TI began with Heinrich Stadial 1 (HS1, ~18-14.7 ka) when part of the Laurentide and Eurasian ice-sheets disintegrated (Dyke, 2004; Hughes et al., 2016), draining freshwater to the North Atlantic, that may have contributed to the observed weakening of NADW formation (e.g. McManus et al., 2004; Gherardi et al., 2009; Thornalley et al., 2011; Ng et al., 2018; Ivanovic et al., 2018). HS1 was charac-

terised by cold and dry conditions in the North Atlantic, over Greenland and Europe (Tzedakis et al., 2013; Buizert et al., 2014;

Martrat et al., 2014). Antarctic temperature and $CO_2$ concentration rose during this period. Paleoproxy records and modelling studies suggest that the Intertropical Convergence Zone (ITCZ) shifted southward (e.g. Arz et al., 1998; Chiang and Bitz, 2005; Timmermann et al., 2005; Stouffer et al., 2007; McGee et al., 2014), thus leading to dry conditions in most of the Northern tropics, including the northern part of South America (Peterson et al., 2000; Deplazes et al., 2013) and the Sahel (Mulitza et al.,

2008; Niedermeyer et al., 2009). Chinese speleothem records also indicate a weak East Asian summer monsoon activity during HS1 (Wang et al., 2001; Cheng et al., 2009).

    The abrupt NADW resumption at ∼14.7 ka, imprinted by a sharp atmospheric $CH_4$ concentration increase (Loulergue et al., 2008) (Fig. 1), led to a warm period in the North Atlantic region, the Bølling-Allerød (Liu et al., 2009; Buizert et al., 2014). This period of North Atlantic warming coincides with (and may have triggered part of) the period of fastest sea-level rise,

Meltwater Pulse-1A (MWP-1A) (Gregoire et al., 2016), during which sea-level rose by 14-18 m in less than 500 years, starting at ∼14.5 ka (e.g. Deschamps et al., 2012; Lambeck et al., 2014). At high southern latitudes, the gradual deglacial warming was interrupted by the Antarctic Cold Reversal (ACR, ∼14.5 - 12.8 ka) (Jouzel et al., 1995, 2007; Pedro et al., 2016), which was also coincident with a pause in the deglacial atmospheric $CO_2$ increase (e.g. Marcott et al., 2014). The ACR could be the result of enhanced northward heat transport from the Southern Hemisphere due to the strong NADW resumption occurring

during the Bølling-Allerød (Pedro et al., 2016), or from a meltwater pulse originating from the Antarctic ice sheet at the time of MWP-1A (Weaver et al., 2003; Menviel et al., 2011; Weber et al., 2014; Golledge et al., 2014).

    A return to stadial conditions over Greenland, Europe and the North Atlantic occurred during the Younger Dryas (∼12.8-11.7 ka, Fig. 1) (Alley, 2000). This event likely resulted from a combination of processes (Renssen et al., 2015), possibly including a weakening of NADW formation resulting from an increase in meltwater discharge into the Arctic Ocean (Tarasov and Peltier,

2005; Murton et al., 2010; Keigwin et al., 2018), or melting of the Fennoscandian ice-sheet (Muschitiello et al., 2015), and an altered atmospheric circulation due to a minimum in solar activity (Renssen et al., 2000). While the Barbados coral record suggests that a second phase of rapid sea-level rise occurred at about 11.3 ka (MWP-1B) (e.g. Bard et al., 1990), data from Tahiti boreholes (Bard et al., 2010) and from a compilation of sea-level data (Lambeck et al., 2014) find no evidence of a particularly rapid sea-level rise during that period.

The penultimate deglaciation (∼138-128 ka, referred here as TII for Termination II), which represents the transition between the penultimate glacial period (MIS 6, also referred to as Late Saalian, 160-140 ka) and the LIG (also referred to as MIS 5e in marine sediment cores) (Govin et al., 2015), has received less attention. MIS 6 was characterised by an atmospheric $CO_2$ content of ∼195 ppm (Lüthi et al., 2008), and a significantly different extent of NH ice-sheets compared to the LGM (Dyke et al., 2002; Svendsen et al., 2004; Lambeck et al., 2006; Ehlers et al., 2011; Margari et al., 2014). Compared to ∼130 m

lower or more during the LGM than during present-day (Austermann et al., 2013; Lambeck et al., 2014), the eustatic sea-level during MIS 6 is estimated at ∼90-100 m lower than present-day (Rabineau et al., 2006; Grant et al., 2012; Rohling et al., 2017), with a relatively large uncertainty range (Rohling et al., 2017). The LIG also bears significant differences to the Holocene. Sea-level was ∼6 to 9 m higher during the LIG than today, thus implying a significant ice-mass loss from both the Greenland and Antarctic ice-sheets (e.g. Dutton et al., 2015). In addition, compared to pre-industrial times, high-latitude SST and Greenland

surface temperatures were respectively ≥1°C and 3 to 11°C greater during the LIG (e.g. Landais et al., 2016; Capron et al.,



2017; Hoffman et al., 2017). Considering the transient nature of the Earth system, a better understanding of TII could thus improve our knowledge of the processes that led to continental ice-mass loss during the LIG.

Recent work (Masson-Delmotte et al., 2011; Landais et al., 2013; Govin et al., 2015) also depicted a sequence of events over TII that contrasts with the one across TI. Paleo-proxy records indicate that the disintegration of NH ice-sheets induced a

~80 m sea-level rise (Grant et al., 2014) between 135 and 129 ka, which could have caused Heinrich Stadial 11 (HS11) (e.g. Heinrich, 1988; Oppo et al., 2006; Skinner and Shackleton, 2006; Govin et al., 2015). HS11 was characterised by weak NADW formation (Oppo et al., 2006; Böhm et al., 2015; Deaney et al., 2017), cold and dry conditions in the North Atlantic region (Drysdale et al., 2009; Martrat et al., 2014; Marino et al., 2015), and gradually warmer conditions over Antarctica (Jouzel et al., 2007), associated with a sustained atmospheric $CO_2$ increase of ~60 ppm (Landais et al., 2013) (Figs. 1, 2). Increasing evi-

dence of sub-millennial scale climate changes at high and low latitudes during HS11 (e.g. Martrat et al., 2014) prompts the need to refine the sequence of events across TII. However, this is challenging as (i) climatic reconstructions over TII are still scarce and most records have insufficient resolution to allow identification of centennial- to millennial-scale climatic variability; and (ii) it is difficult to establish robust absolute and relative chronologies for most paleoclimatic records across this time interval (Govin et al., 2015).

While our knowledge of the processes and feedbacks occurring during deglaciations has significantly improved over the last two decades (e.g. Cheng et al., 2009; Shakun et al., 2012; Abe-Ouchi et al., 2013; Landais et al., 2013; Cheng et al., 2016), many unknowns remain. For example, our understanding of the precise roles of atmospheric and oceanic processes in leading to the waning of glacial continental ice-sheets during deglaciations is still incomplete. It is also crucial to comprehend the subsequent impacts of continental ice-sheets disintegration on the oceanic circulation and thus the climate and carbon-cycle

system.

Numerical simulations performed with climate models provide a dynamical framework to understand the response of the Earth system to external (i.e. insolation) and internal (e.g., albedo, GHGs) forcings that culminate in deglaciations. Atmospheric and oceanic teleconnections associated with millennial-scale variability can also be studied in detail. Model-paleoclimate proxy comparisons, including snapshot experiments at 130 ka and at 126 ka, suggest that the inclusion of freshwater forcing in the

North Atlantic due to the melting of NH ice sheets could explain the relatively cold conditions in the North Atlantic and warm conditions in the Southern Ocean during these time periods (Govin et al., 2012; Stone et al., 2016). However, snapshot experiments assume that the climate state is in near-equilibrium, and because relatively rapid and large changes in both internal and external forcings occur during deglaciations, transient simulations (i.e. numerical simulations with time-varying boundary conditions) are needed. These simulations also allow a more robust paleodata/modelling comparison, thus enabling the refinement

of the sequence of events.

Transient numerical simulations have already been performed for TI (Liu et al., 2009; Menviel et al., 2011; Roche et al., 2011; Gregoire et al., 2012; He et al., 2013; Otto-Bliesner et al., 2014) and provide a dynamical framework to further our understanding of the climate-change drivers, teleconnections and feedbacks inherent in the Earth system. Transient simulations covering the period 135 to 115 ka have also been performed with a range of models to understand the impact of surface

boundary conditions and freshwater fluxes on the LIG (Bakker et al., 2013; Loutre et al., 2014; Goelzer et al., 2016). The need





for transient simulations is now recognised and a protocol to perform transient experiments of TI as part of PMIP4 has been recently established (Ivanovic et al., 2016). However, to further our understanding of the processes at play during terminations, including the role of millennial-scale climate change, other terminations should be studied in detail. It is also useful to evaluate the influence of terminations on the climate of the following interglacial. We thus propose to extend the PMIP4 working group

on the last deglaciation to include the penultimate deglaciation and thus create a DeglaMIP working group. This effort will complement the TI experiments of PMIP4 (Ivanovic et al., 2016), allowing an evaluation of the similarities and differences in the climate system response during TI and TII in concert with paleoclimate records.

Here we present a protocol to perform transient numerical simulations of TII from 140 to 127 ka in order to provide a link with the PMIP4 transient LIG experiment (127 to 121 ka) (Otto-Bliesner et al., 2017). After a description of changes in

insolation, GHGs, continental ice-sheets, sea-level and oceanic circulation occurring during TII, we present a framework to perform transient simulations of TII, as well as a selection of key paleoclimate and paleoenvironmental records to be used for model/data comparisons.

## 2   Insolation

The orbital parameters (eccentricity, obliquity, and longitude of perihelion) should be time evolving and set following Berger

(1978). This external forcing affects the seasonal and latitudinal distribution, as well as the magnitude of solar energy received at the top of the atmosphere and, in the case of obliquity (Earth's axial tilt), the annual mean insolation at any given latitude with opposite, but small, effects at low and high latitudes. Eccentricity is high during the entire TII period, ranging from 0.033 at 140 ka to 0.041 at 127 ka, and is significantly higher than during TI (Fig. 1b; ∼0.019 at the LGM to 0.020 at 14 ka), and than the present value of 0.0167. Obliquity peaks at 131 ka; the degree of tilt is similar between TII and TI. Perihelion occurs

near the NH winter solstice at 140 ka, shifting to near the NH summer solstice by 127 ka.

Although the overall trends in summer solstice insolation anomalies, as compared to the mean of the last 1000 years, evolve similarly in TII and TI, the magnitudes of the maximum positive summer anomalies in the NH and the minimum negative summer anomalies in the Southern Hemisphere are much greater in TII, when eccentricity is higher, than during TI (Fig. 1a). At 65°N, peak summer anomalies of more than 70 W/m$^2$ occur at 128 ka. In contrast, during TI, the 65°N summer solstice

anomalies peak at 11 ka, with anomalies of ∼50 W/m$^2$. Similarly, 65°S summer solstice negative anomalies are close to -40 W/m$^2$ at 127 ka, but only about -20 W/m$^2$ at 10 ka. In addition, the rates of change of summer solstice anomalies are greater from 140 to 127 ka than from 21 to 8 ka.

Given the clear differences between the solar forcing of TI and TII, comparing the two transient deglacial simulations will provide valuable information on the underlying mechanisms and Earth system feedbacks. The solar constant should be set to

1360.7 W/m$^2$, consistent with the CMIP6-PMIP4 piControl and lig127k simulations (Otto-Bliesner et al., 2017) as well as the PMIP4 transient climate simulation of the last deglaciation (Ivanovic et al., 2016).



## 3  Greenhouse gases

GHG records are available solely from Antarctic ice cores across the time interval 140-127 ka (Fig. 2). LIG GHG records from the NEEM and other Greenland ice cores are affected by stratigraphic disturbances and in-situ $CO_2$, $CH_4$ and $N_2O$ production (e.g. Tschumi and Stauffer, 2000; NEEM community members, 2013). The NGRIP ice core provides a continuous and reliable

$CH_4$ record but it only extends back to ∼123 ka (North Greenland Ice Core Project members, 2004). After a brief description of existing atmospheric $CO_2$, $CH_4$ and $N_2O$ records (below), we recommend using the latest GHG records measured on the EPICA Dome C (EDC) ice core (Loulergue et al., 2008; Schilt et al., 2010; Schneider et al., 2013). They should be displayed on the commonly-used AICC2012 gas age scale (Bazin et al., 2013; Veres et al., 2013). Note that this time scale is associated with an average $1\sigma$ absolute error of ∼2 kyr between 140 and 127 ka.

Atmospheric $CO_2$ concentration records have been measured on the EDC and TALDICE ice cores (Fig. 2). The EDC records from Lourantou et al. (2010) and Schneider et al. (2013), characterised by a temporal resolution of ∼0.5 and 0.8 kyr respectively, agree well overall. The Schneider et al. (2013) dataset depicts a long-term $CO_2$ increase starting at ∼137.8 ka and ending at ∼128.5 ka with a centennial-scale $CO_2$ rise above the subsequent LIG $CO_2$ values, also referred to as an "overshoot". The $CO_2$ overshoot is smaller in the Schneider et al. (2013) dataset compared to a similar feature measured in Lourantou et al.

(2010): while the former displays a relatively constant $CO_2$ concentration of ∼275 ppm between 128 and 126 ka, the latter shows a $CO_2$ decrease from 280 to 265 ppm between 128 and 126 ka. The offsets between $CO_2$ records from the same EDC core are likely related to the different air extraction techniques used in the two studies (Schneider et al., 2013).

Consistent with the recommendation made for the PMIP4 transient simulations of the last deglaciation (Ivanovic et al., 2016), we recommend the use of the EDC $CO_2$ record from Schneider et al. (2013) displayed on the AICC2012 timescale as

provided by Bereiter et al. (2015).

Atmospheric $CH_4$ concentration records from Vostok, EDML, EDC and TALDICE agree well within the gas-age uncertainties attached to each core (Fig. 2). They illustrate a slow rise from ∼390 to 540 ppbv between ∼137 ka and 129 ka that is followed by an abrupt increase of ∼200 ppbv reaching maximum LIG values at ∼128.5 ka. Because $CH_4$ sources are located mostly in the NH, an interpolar concentration difference (IPD) between Greenland and Antarctic $CH_4$ records exists.

For instance, an IPD of ∼14 ppbv, ∼34 ppbv and ∼43 ppbv is reported during the LGM, Heinrich Stadial 1 and the Bølling warming respectively (Baumgartner et al., 2012). However, without reliable $CH_4$ records from Greenland ice cores, it remains challenging to estimate the evolution of the IPD across TII. Thus, for the atmospheric $CH_4$ forcing, we recommend using the EDC record, with a temporal resolution of ∼0.3 kyr (Loulergue et al., 2008), recognising that the values may be 1-3% lower than the actual global average.

Atmospheric TALDICE, EDML and EDC $N_2O$ records are available between 134.5 and 127 ka (Fig. 2) (Schilt et al., 2010; Flückiger et al., 2002). From 134.5 to 128 ka, $N_2O$ levels increase from ∼220 to 270 ppbv. Following a short decrease until ∼127 ka, $N_2O$ concentrations stabilise afterwards. No reliable atmospheric $N_2O$ concentrations are available beyond 134 ka as $N_2O$ concentrations measured in the air trapped in ice from the penultimate glacial maximum are affected by in-situ production related to microbial activity (Schilt et al., 2010). During the LGM (considered here as the time interval 26-21 ka),



the average $N_2O$ level was $\sim$201 ppbv. Assuming the LGM is an analogue for the penultimate glacial maximum, we propose to use a constant $N_2O$ level between 140 and 134.5 ka taken as the LGM value of 201 ppbv. Even though it is possible that the high value at $\sim$134ka could relate to in-situ artifacts that affect inland ice cores, we recommend using the EDC atmospheric $N_2O$ record with a temporal resolution of $\sim$0.6 kyr (Flückiger et al., 2002) (Fig. 2). We note that this choice should not

significantly impact the results of the transient simulations given the relatively small radiative forcing values of $N_2O$ ($\sim$0.15 $W/m^2$) compared to $CH_4$ ($\sim$0.32 $W/m^2$) and $CO_2$ ($\sim$2.1 $W/m^2$) across TII (Köhler et al., 2017).

## 4   Continental ice-sheets

Changes in continental ice-sheets during TII will significantly impact the climate system through their albedo, which will directly affect the radiative balance (e.g. He et al., 2013). Changes in continental ice-sheets geometry can also significantly im-

pact atmospheric dynamics (e.g. Zhang et al., 2014; Gong et al., 2015). Transient simulations of TII will thus need to be forced by the 3-dimensional and time-varying evolution of continental ice-sheets, that is currently only available from numerical simulations. However, simulating the evolution of continental ice-sheets across TII is associated with large uncertainties, due to the climate forcing of the ice-sheet models and poorly constrained non-linearities within the ice-sheet system. Glacial geological data are also available to constrain continental ice-sheet evolutions and can thus provide an estimate of the uncertainties

associated with the numerical ice-sheet evolutions. In this section, we describe the available numerical ice-sheet evolutions to use as a forcing of the transient simulations of TII. We further compare the results of these simulations with existing glacial geological constraints.

### 4.1   Combined ice-sheet forcing

To facilitate the transient simulations of TII, we are providing a combined ice-sheet forcing (available on the PMIP4 portal),

in which all different ice-sheets are merged. The simulated NH ice sheet evolution, described in Section 4.2 (Abe-Ouchi et al., 2013), is merged with the simulated 240 kyr Greenland and the Antarctic (Briggs et al., 2014) evolutions described in Sections 4.3 and 4.4, respectively. The resolution of the merged ice-sheet file is 1° longitude by 0.5° latitude. From the LIG onward, the combined ice-sheet evolution, referred as GLAC1-D in PMIP4, is used (e.g. Ivanovic et al., 2016). GLAC-1D includes the Greenland and Antarctic ice-sheets components described in section 4.3 and 4.4 (Briggs et al., 2014), the North American ice-

sheet simulation described in Tarasov et al. (2012) and the Eurasian ice-sheet simulation described in Tarasov (2014). The ice sheet thickness from these simulations are run through a sea-level solver using the VM5a (Peltier and Drummond, 2008) Earth rheology to extract a gravitationally self-consistent topography (thus the need for the full 240 kyr ice sheet evolution). The surface topography is then run through a global surface drainage solver (using the algorithm described in Tarasov and Peltier, 2006) to extract the surface drainage pointer evolution.





## 4.2 North American and Eurasian ice-sheets

The evolution of NH ice-sheets during TII is given by a numerical simulation performed with the thermo-mechanically coupled shallow-ice-sheet model IcIES (Ice sheet model for Integrated Earth system Studies) with an original resolution of 1° by 1° in horizontal and 26 vertical levels (Abe-Ouchi et al., 2007) (Fig. 3). This ice-sheet model was driven by climatic changes obtained from the MIROC GCM (Abe-Ouchi et al., 2013), which was forced by changes in insolation and atmospheric $CO_2$ concentration. In global agreement with glacial geological constraints (Dyke et al., 2002; Svendsen et al., 2004; Curry et al., 2011; Syverson and Colgan, 2011) and numerical simulations of NH ice-sheets evolution (Tarasov et al., 2012; Abe-Ouchi et al., 2013; Peltier et al., 2015; Colleoni et al., 2016), the simulated extent and volume of the North American ice-sheet was smaller during MIS 6 than MIS 2 (Fig. 3).

In Eurasia, MIS 6 recorded the most extensive glaciation of the last 400 ka (Hughes and Gibbard, 2018). The maximum extent of the Fennoscandian ice-sheet probably occurred at ∼160 ka, when it extended into central Netherlands, Germany, and the Russian Plain (Margari et al., 2010; Ehlers et al., 2011; Hughes and Gibbard, 2018). This was followed by a partial melting of the Fennoscandian ice-sheet, peaking between ∼157-154 ka, and a readvance after 150 ka (Margari et al., 2010; Hughes and Gibbard, 2018). The maximum extent of the NH ice-sheets probably occurred at the end of MIS 6 (Margari et al., 2014; Head and Gibbard, 2015), due to ice-sheet expansion in Russia, Siberia (Astakhov et al., 2016) and in North America (e.g. Curry et al., 2011; Syverson and Colgan, 2011). Glacial geological constraints (e.g. Astakhov, 2004; Svendsen et al., 2004) indeed suggest that the Barents-Kara ice-sheet extended further during MIS 6 than in MIS 2. The simulated Eurasian ice-sheet is in general agreement with the reconstruction of Lambeck et al. (2006), with a dome reaching 3000 m over the Kara Sea during MIS 6 that subsequently disintegrated across TII. However, the extent and volume of the simulated Eurasian ice-sheet might be underestimated since it is smaller at MIS 6 than MIS 2, whereas reconstructions suggest it should be larger at MIS 6 (Lambeck et al., 2006; Rohling et al., 2017).

Rohling et al. (2017) further suggest that the ice volume was almost equally distributed between Eurasia and North America at MIS 6, with a 33 to 53 m global mean sea level equivalent (sle) contribution from the Eurasian ice-sheet and 39-59 m from North America, whereas the ice-sheet simulation produces a ∼24 m sle contribution from Eurasia. Thus, the volume of the North American ice-sheet may also be overestimated.

In the ice-sheet simulation, NH ice-mass loss follows closely the boreal summer insolation and occurs mostly between ∼134 and 127 ka (Fig. 4), with two peaks of glacial meltwater release at ∼131 ka and 128 ka. By 132 ka, the Eurasian ice-sheet has decreased significantly and the southern and western flanks of the North American ice-sheet have disintegrated (Fig. 3). Another significant retreat of the North American ice-sheet occurs between 132 and 128 ka, at which point it is mostly restricted to the north of the Hudson Bay.

## 4.3 Greenland ice-sheet

The Greenland model uses an updated version (GSM.G7.31.18) of the Glacial Systems Model (e.g. Tarasov et al., 2012) run at grid resolution of 0.5° longitude by 0.25° latitude. The model has been upgraded to hybrid shallow-ice and shallow-shelf



physics, with ice dynamical core from Pollard and DeConto (2012) and includes: a 4 km deep permafrost resolving bed thermal component (Tarasov and Peltier, 2007), visco-elastic bedrock response with global ice sheet and sea-level loading, sub-shelf melt, parametrizations for subgrid mass-balance and ice flow (Morzadec et al., 2015), and updated parametrizations for surface mass-balance and ice calving.

Model runs start at 240 ka with present-day ice and with an ice and bed temperature field from the end of a previous 240-kyr model run. The model is then forced from 240 ka until 0 ka, with a climate forcing that is partly glacial index based, using a composite of a glaciological inversion of the GISP II regional temperature change (for the last 40 kyr) and the synthetic Greenland $\delta^{18}$O curve that was deduced from the Antarctic EDC isotopic record assuming a thermal bipolar seesaw pattern (Barker et al., 2011). The climate forcing also includes 2-way coupled 2D energy balance climate model (Tarasov and Peltier,
1997) to capture radiative changes.

Greenland ice sheet model runs are scored against a large set of constraints including relative sea level (RSL), proximity to present-day ice-surface topography, present-day observed basal temperatures from various ice cores, time of deglaciation of Nares Strait, and the location of the present-day summit. The simulation presented here (G9175) is a least misfit model from a preliminary exploratory ensemble. The last 20 kyr of the run is thus critical as this represents the time period with most of the
data constraints for Greenland.

This simulation suggests no significant change in Greenland ice-mass until ∼134 ka (Fig. 4), followed by a small ice-mass loss, mostly from floating ice, between 134 and 130 ka. In this simulation, the main phase of Greenland deglaciation occurs between 130 and 127 ka, during which Greenland loses its glacial non-floating ice volume (∼2.9 m sle), but also loses an additional 1.5 m sle, compared to the pre-industrial Greenland configuration. As shown in Figure 5, the extent and height
of the Greenland ice-sheet is significantly smaller at 128 ka than at 132 ka. Greenland ice-mass loss is particularly evident on its western side, with a part of southwestern Greenland being ice-free. To a first order, the simulated disintegration of the Greenland ice-sheet follows the increase in boreal summer insolation and in atmospheric $CO_2$ (Fig. 1).

The main phase of the Greenland ice-sheet retreat in this simulation is globally in agreement with proxy records, which suggest significant runoff in the Labrador Sea at ∼130 ka and at ∼127 ka (e.g. Carlson and Winsor, 2012). However the
simulated Greenland ice-sheet disintegration could be too rapid as paleoproxy records suggest significant meltwater discharge from the Greenland ice-sheet throughout the LIG (e.g. Carlson and Winsor, 2012). In addition other model simulations suggest a maximum sea-level contribution from Greenland at ∼123-121 ka (Yau et al., 2016; Bradley et al., 2018), in agreement with the timing of the LIG minimum elevation at the Greenland NEEM location. This minimum elevation estimate was reconstructed from total air and water isotopic records measured on the deep ice core drilled at that site (NEEM community members, 2013).
This maximum sea level Greenland contribution is also coherent with the NH ice-sheets contribution to global sea-level during the LIG (Kopp et al., 2009).

## 4.4 Antarctic ice-sheet

The Antarctic model configuration is largely that of Briggs et al. (2013, 2014): a hybrid of the Penn State University ice-sheet model (Pollard and DeConto, 2012) and the GSM. Simulations are run with a 40-km grid resolution using the LR04





benthic $\delta^{18}O$ stack (Lisiecki and Raymo, 2005) for sea-level forcing. The climate forcing is a parametric mix of an index based approach (using the EDC $\delta D$ record of Jouzel et al. (2007)) and one based on orbital forcing, as detailed by Briggs et al. (2013). The parameter vector (nn4041 from Briggs et al., 2014) that gave the best fit to constraints (GSM-A) in the large ensemble analysis is used. Two changes are imposed on the model to partially rectify an inadequate LIG sea-level contribution.

First, SST dependence is added to the sub-shelf melt model. Second, to compensate for inadequate LIG warming, where SSTs are above present-day values, they are then given a minimum value of 3°C (i.e. SST=MAX(SST,3.0°C)). Even so, the Antarctic contribution to the LIG high-stand is only 1.4 m sle and is therefore inadequate given current inferences (as well as constraints on contributions from Greenland and steric effects) (e.g. Kopp et al., 2009).

During TII, the simulation suggests a continuous Antarctic ice-sheet discharge, with a glacial ice-mass loss of ∼12.5 m sle

between 140 and 131 ka, followed by an additional 1.4 m sle between 131 and 130 ka (Fig. 4). In this simulation, the West Antarctic Ice Sheet loses significant ice-mass between 140 and 136 ka (Fig. 6), with a retreat of the grounding line over the Ross Sea as well as ice-mass loss in the Weddell Sea, on the Antarctic Peninsula and in the Amundsen Sea sector. By 132 ka, the grounding line has completely retreated over the Ross Sea and has retreated significantly over the Weddell Sea.

## 5   Sea-level

Direct evidence for constraining the evolution of the global sea level during the time interval 140-127 ka remains sparse. Although the LR04 benthic $\delta^{18}O$ stack (Lisiecki and Raymo, 2005) is sometimes used to approximate sea-level change on glacial-interglacial timescales, in the case of TII the timing of the LR04 benthic $\delta^{18}O$ stack is fixed by reference to a handful of U-series coral dates from Huon Peninsula with relatively high analytical uncertainties and questionable preservation (Bard et al., 1990; Stein et al., 1993). Tying the MIS 5e peak to the average age of these coral dates results in a benthic $\delta^{18}O$

minimum that is roughly centred on the main phase of coral growth during this interglacial period (122 - 129 ka) (Stirling et al., 1998) rather than having the onset of the interglacial aligned with the timing of the onset of the sea level high-stand at far-field sites (∼129 ka) (e.g. Stirling et al., 1998; Dutton et al., 2015).

Here, we seek to provide an improved reconstruction of sea level across TII by examining available RSL records. Information on the timing and magnitude of the changes across this time interval is provided by three RSL records (Fig. 7):

*i)* A RSL record from the Red Sea (Grant et al., 2012), that is deduced from the planktic foraminifera $\delta^{18}O$ measured on sediment cores retrieved in this evaporative marginal sea. This record is transformed into a RSL signal by using hydraulic models that constrain the salinity of surface waters as a function of sea level. The Red Sea record provides the only continuous profile of RSL across our interval of interest;

*ii)* RSL data from the U-series dates and elevations of the submerged coral reefs of Tahiti (Thomas et al., 2009), and

*iii)* RSL data derived from U-series dates and elevations of uplifted coral terraces of Huon Peninsula, Papua New Guinea (Esat et al., 1999).

Providing a robust age model for sediment records from MIS 6 to the LIG is not straightforward (e.g. Govin et al., 2015) and over time, several age models have been proposed for the Red Sea RSL record (e.g. Siddall et al., 2003; Rohling et al., 2009;



Grant et al., 2012). The latest chronology is based on climatic alignment of the Red Sea RSL record to eastern Mediterranean planktic foraminifera $\delta^{18}$O records, which are in turn aligned onto the absolutely-dated Soreq cave speleothem $\delta^{18}$O record (Grant et al., 2012). While the absolute ages of the speleothem record have the potential to provide a more robust age model (both in terms of accuracy and precision), the application of these dates to the Red Sea sea level reconstruction hinges on the

assumption that the tie points between the Red Sea and eastern Mediterranean records have been correctly assigned, and that the intervals between these tie points can be linearly extrapolated.

The Tahiti and Huon Peninsula corals are associated with absolute radiometric dates (using U-series geochronology). For the purpose of this study, all of the U-series ages have been recalculated to normalize them with the same set of decay constants for $^{234}$U and $^{230}$Th (Cheng et al., 2013) (Tables S3 and S4), using the methodology described by Hibbert et al. (2016). Note that

the array of data from Huon Peninsula suggest post-depositional alteration (open-system behaviour of the U-series isotopes) that complicates a precise age interpretation (Fig. 7).

The Red Sea time series published by Grant et al. (2012) depicts that, after a RSL low stand of about -100 m relative to present between 145 and 141 ka, a brief pulse of at least ∼25 m sea-level rise, based on the smoothed record (or up to ∼50 m based on the unsmoothed time series), occurred between ∼141 and 138 ka (identified as MWP-2A in Marino et al. (2015), Fig.

7a). This pulse was followed by a slight sea level fall (∼10 m in the smoothed record) between ∼139 and 138 ka. Finally, a more significant pulse of ∼70 m in RSL rise (MWP-2B) is inferred between 135 and 130 ka. The period between the ephemeral pulse of sea-level rise at the beginning of TII (MWP-2A) and the second prolonged pulse (MWP-2B/HS11), has sometimes been referred to as the TII sea-level reversal (Siddall et al., 2006).

The coral RSL data from Huon Peninsula and Tahiti independently provide additional evidence for an ephemeral reversal

in sea level rise occurring during the penultimate deglaciation (Fig. 7). In the case of Tahiti, sedimentary evidence for the superposition of shallow and deeper water facies led to the interpretation that there was an ephemeral deepening (sea-level rise) followed by a return to shallower water conditions (sea-level fall or stabilization) (Thomas et al., 2009). The Tahiti data provide bounding ages on the timing of this sea level rise pulse, with ages of corals that grew at 135.0 ka (in 0-6 m water depth) and 133.5 (±1) ka (0-25 m water depth). In between these shallower facies, there is a deeper water facies (≥20 m paleowater

depth), but there are no reliable ages within this interval of the core (Thomas et al., 2009). This observation, based on changes in both the lithofacies and benthic foraminiferal assemblage, is interpreted as a pulse of sea-level rise in between about 135.0 and 133.5 ka (Fujita et al., 2010). A similar sea level oscillation has also been interpreted based on the stratigraphy as well as the age and paleowater depth reconstruction at Huon Peninsula (Esat et al., 1999). The absolute timing of coral growth is only loosely constrained at this site due to open-system behaviour of the U-series isotopes (as reflected by the scatter in ages

of corals collected in Aladdin's cave, ∼134 to 126 ka, Fig. 7) (Esat et al., 1999). Indeed, the corals from Terrace VII have ages (with high uncertainty) ranging from about 137 to 134.5 ka and the corals from the cave have a wide range of ages, from 134.1 to 125.9 ka (more details in Esat et al. (1999)). Given that the younger end of this age range is clearly within the MIS 5e sea-level highstand (e.g. Stirling et al., 1998), it is more likely that the older end of this diagenetic array of data from Aladdin's cave is a better approximation for a the primary age (i.e. it is closer to the unaltered end member). Despite these diagenetic





concerns, the agreement in the timing of this TII sea level reversal (MWP-2A) in Tahiti and Huon Peninsula is striking (Fig. 7b).

We note that, in comparison, the coral records suggest that MWP-2A occurs considerably later (i.e. ∼135-134 ka) than the Red Sea RSL reconstruction when displayed on the chronology from Grant et al. (2012) (∼141-138 ka). Such a mismatch is

likely to be related to dating uncertainties associated with the current Red Sea RSL age scale. Hence, we propose that the tie point introduced in Grant et al. (2012) to stretch the depth scale across this interval, should be modified such that the timing of MWP-2A is consistent with the absolute ages provided by the Tahiti and Huon Peninsula coral data. We note that reassigning the tie points across this interval (Table S5), where tie points are placed at the beginning and end of MWP-2A (as defined by the coral data), results in a sea-level reconstruction that more closely approximates a linear age-depth model (Fig. 7a). This revised

age model for the Red Sea RSL is adopted as our preferred reconstruction for sea-level change during TII. This reconstruction also compresses the total duration of the sea-level rise during the entirety of the TII transition, which has implications for the freshwater forcing in the NH and for making analogies between TII and TI.

This reconstruction is still subject to debate given the limits of the datasets. For example, considerable uncertainties remain with the magnitude of the sea-level pulse during MWP-2A because some of the corals cover a wide range of paleowater depth

(0 to 6 m for the pre-MWP-2A Tahiti corals, ≥20 meters for the Tahiti corals during MWP-2A, 0-25 m for the post-MWP-2A Tahiti corals and 0 to 20 m for the Aladdin's Cave corals). Despite these uncertainties in the absolute position of sea level, the relative sea-level changes for each site clearly demonstrate an ephemeral deepening during meltwater pulse MWP-2A in both cases.

Glacial isostatic adjustment to the deterioration of the MIS 6 ice sheets will also differentially affect Tahiti and Huon Penin-

sula, which precludes a direct comparison of the magnitude of sea-level change between these sites or a direct interpretation of global mean sea-level change in the absence of modelling. Because the changes in global mean sea level are rapid across the penultimate deglaciation, the eustatic signal is likely dominant, leading to a timing of the rapid changes that is similar between local RSL and global mean sea-level reconstructions. Still, the rate of change may be different between sites due to local differences in the magnitude of sea-level change. Based on the revised chronology for the Red Sea RSL and on the coral

constraints, MWP-2A starts at ∼137 ka, while MWP-2B starts at ∼133 ka (Fig. 7, Table 3).

Finally, far-field coral data from the Seychelles and Western Australia, that have been corrected for the glacial isostatic adjustment (e.g. Dutton et al., 2015), indicate that global mean sea level passed the position of modern sea level at about 129 ka (Fig. 7). The evolution of sea level during the LIG high-stand is still debated and may have included some meter-scale sea-level oscillations, but by at least some accounts, it is thought to have risen a few meters between 129 and 122 ka (e.g.

Kopp et al., 2009; Dutton et al., 2015). So while the timing of peak sea level may have occurred later in the interglacial (∼122 ka), the onset of the highstand (∼129 ka) could represent an inflexion point in the rate of sea-level change coming out of the rapid deglaciation and into the interglacial.

Overall, eustatic sea-level reconstructions based on paleodata and continental ice-sheets simulations (Section 4) are consistent. However, the amplitude of the eustatic sea level change across TII estimated from the Red Sea reconstructions is ∼10 m

smaller than the combined ice-sheets simulations (Fig. 4e). In addition, the sea-level data suggest a small sea-level increase




at ∼140 ka, which is not present in the ice-sheet simulations. Both suggest that the main phase of sea-level rise/continental ice-sheets disintegration initiates at ∼134 ka, even if the overall magnitude is larger in the ice-sheet simulation, but with a lower rate of change than in the Red Sea reconstructions. However, it is worth keeping in mind that both the sea level data- and ice-sheet model-based approaches are associated with large uncertainties regarding the exact timing and amplitude of global

sea-level changes across TII.

## 6    Recommendations for transient simulations of TII

### 6.1    Equilibrium spin-up at 140 ka

If a LGM run is already available, then it is suggested to initialise the 140 ka spin up from the climate fields and ocean state produced by the LGM equilibrium run. Starting from a LGM state may minimize the duration of the spin-up, as it

should shorten the time to reach equilibrium in the deep ocean (Zhang et al., 2013). If starting from a pre-industrial set up, it is suggested to follow the PMIP4 LGM protocol (Kageyama et al., 2017) to set up the 140 ka state, but using the 140 ka boundary conditions described below instead of the respective 21 ka boundary conditions.

The model should be forced with 140 ka background conditions (Table 1), including appropriate orbital parameters, GHG concentrations as averaged over the interval 141-139 ka (195 ppm $CO_2$, 387 ppb $CH_4$, and 201 ppb $N_2O$), as well as the NH

and Antarctic ice-sheets' extent, topography and associated albedo (as described above). Forcing files describing the evolution of NH and Antarctic ice-sheets, as simulated by ice-sheets models (Abe-Ouchi et al., 2013; Briggs et al., 2014) (Section 4), are provided in the associated data repository. They include the evolution of the ice-mask, as well as surface and bedrock elevations. Kageyama et al. (2017) provide guidelines for computing land fraction, land-ice fraction and orography from the ice-sheet reconstruction datasets. The details of these forcings and the approach taken to compute them will ultimately depend

on each model resolution and restrictions.

The large glacial ice-sheets of MIS 6 impacted sea level and the land-sea mask. It would be best to modify the land-sea mask resulting from ice-sheet changes. Depending on the resolution of the model, this might not be a crucial parameter, except for some bathymetry and land-sea mask features, which have particular importance for ocean circulation. For example, the Bering Strait, which is 40 to 50 m deep, exerts a significant control on NADW and North Pacific Intermediate Water

formation (e.g Okazaki et al., 2010; Hu et al., 2012). We therefore recommend to close the Bering Strait during the 140 ka spin up. Following the recommendations for the PMIP4 LGM equilibrium run (Kageyama et al., 2017), the land-sea mask should include the exposure of the Sahul and Sunda shelves in the Indo-Australian region, as well as closure of the strait between the Mediterranean Sea and the Black Sea.

To account for the maximum MIS 6 ice-sheet expansions, and associated global sea-level of ∼100 m below pre-industrial

times, global salinity should be set at +0.85 psu above pre-industrial level. Furthermore, if included in the simulations, the ocean mean $\delta^{18}O$ should be initialized at 1‰ and global mean alkalinity content should be increased by about 80 $\mu$mol/L. Compared to the LGM state, these values are 0.15 psu lower for global salinity, 0.2‰ lower for mean ocean $\delta^{18}O$ and about 16 $\mu$mol/L lower for global alkalinity.



The model should be spun up until near equilibrium is reached. Previous PMIP protocols recommend that the simulations are considered at equilibrium when the trend in globally averaged SST is less than 0.05°C per century and the Atlantic Meridional Overturning Circulation (AMOC) is stable. Marzocchi and Jansen (2017) recently pointed out that the AMOC should be monitored on a centennial timescale to properly assess equilibrium. Zhang et al. (2013) further suggested that the trend in

zonal-mean salinity in the Southern Ocean (south of the winter sea-ice edge) should remain small (less than 0.005 psu per 100 years), especially in the Atlantic sector, to avoid potential transient characteristics in the deep ocean from impacting on AMOC strength. For models including representations of the carbon cycle or dynamic vegetation, the requirement is that the carbon uptake or release by the biosphere is less than 0.01 PgC per year. Similar to the recommendation made in Kageyama et al. (2017), the outputs of at least 100 years of the equilibrated 140 ka spin up should be made available and fully described.

## 6.2   Transient forcings across TII

The main changes in boundary conditions across TII, i.e. insolation, GHG concentrations and continental ice-sheets, have been described in sections 2, 3 and 4, respectively and are summarised in Table 1. For all simulations, methods should be fully documented.

### 6.2.1   Orography, bathymetry, coastlines and rivers

Disintegration of continental ice sheets during TII affected continental topography and ocean bathymetry, and thus coastal outlines and river routing. Therefore, time-varying changes in land-ice fraction, land-sea fraction, topography and bathymetry should be applied. Variations in the ice mask and topography should be updated at the same time. It is up to each group to decide the appropriate time frequency at which to update this forcing. The resolution of the files provided is 500 years, but higher frequency changes obtained through linear interpolations can also be performed to avoid step changes. As mentioned for the

140 ka spin-up regarding changes in land-sea fraction, particular attention should be given to the opening of the Bering Strait and the flooding of the Sunda and Sahul shelves. When possible, these should be varied across TII as ice-sheets disintegrate and sea-level rises. Following the combined ice-sheet history presented here, which includes some GIA adjustment, the Bering Strait might open at ∼127.5 ka. Please note that this is a first estimate, which is associated with large uncertainties.

As changes in the land-sea mask could impact water delivery to the ocean through rivers, it is recommended to check that

river mouths are consistent with the adjusted land-sea mask (Kageyama et al., 2017). If possible and of interest, river networks could also be remapped to take into account the ice-sheet changes. Topographically-self consistent drainage routing maps will be provided.

### 6.2.2   Vegetation, land surface and other forcings

The climatic and ice-sheet changes occurring between 140 and 127 ka will significantly impact the vegetation, and thus also

land albedo, evapo-transpiration and the terrestrial carbon pool. The preferred option would thus be to include a dynamical vegetation model, fully coupled to the atmospheric model. However, care should be taken in regions where an ice-sheet is

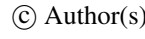



present, as the ice-sheet and its albedo should replace any possible vegetation. If a coupled dynamical vegetation model is not available, then the experiments should be run with prescribed land-surface parameters and with fixed vegetation types and plant physiology outside of the regions covered by a continental ice-sheet, as obtained from the CMIP5 pre-industrial set up (Taylor et al., 2012; Ivanovic et al., 2016). In that case, care should be taken to have a consistent land-surface/vegetation

forcing with the adjusted land-sea mask. Vegetation/land-surface type on a newly emerged land (for example the Sunda shelf) will thus need to be fixed based on interpolations with the nearest grid points.

### 6.2.3 Freshwater forcing

Through their impact on global salinity and ocean circulation, disintegrating ice-sheets can significantly affect the climatic and biogeochemical evolution of the penultimate deglaciation (Oppo et al., 1997; Cheng et al., 2009; Hodell et al., 2009;

Landais et al., 2013; Deaney et al., 2017). In particular, meltwater input in the North Atlantic region could be a significant driver of changes in NADW formation (Loutre et al., 2014; Goelzer et al., 2016; Stone et al., 2016), including the ones associated with HS11 (further discussed in Section 8.2). It is thus strongly recommended to include a carefully designed freshwater scenario when performing transient simulations of TII.

As much as possible, and for all scenarios, meltwater should be added in the appropriate locations to match the evolution

of the ice sheets. A self-consistent paleo surface drainage forcing could also be implemented, which would involve using the provided downslope routing fields to route net grid cell water flux (fwf):

$$fwf = (P-E)_{GCM} - (dH/dt)_{ice-sheet}, \tag{1}$$

with P for precipitation, E for evaporation and $(dH/dt)_{ice-sheet}$ the change in ice-sheet thickness over time as described in the ice-sheet forcing files. Modellers may need to adjust the above to account for land surface model changes in water storage.

Estimates of sea-level changes across TII suggest a global sea-level rise of ∼100 m (Grant et al., 2012, 2014) (Fig. 7) between 140 and 130 ka, due to the disintegration of NH and Antarctic ice sheets. As the Antarctic contribution to sea level is estimated at ∼13 m (Fig. 4), this leaves a ∼87 m contribution from NH ice sheets.

For the NH, three meltwater scenarios are proposed based *i)* on changes in NH ice sheets, as described in section 4 (Fig. 4f, black, fIC) (Abe-Ouchi et al., 2013), *ii)* on global mean sea-level changes with our revised chronology (see section 5, Fig. 4f,

blue, fSL) minus a linear Antarctic contribution of 13 m (Fig. 4d) (Briggs et al., 2014), and *iii)* as derived from North Atlantic and Norwegian Sea IRD records (see also section 8.2, Fig. 4f, red, fIRD).

Estimates of meltwater input to the North Atlantic based on changes in NH ice sheets (Fig. 4b,e,f, black, fIC) (Abe-Ouchi et al., 2013) suggest a sustained (≥0.1 Sv) meltwater flux between ∼133 and 127 ka. This forcing closely follows changes in high northern latitude summer insolation and would probably lead to a significant NADW weakening during HS11, lasting until

about 125.5 ka depending on the model's sensitivity.

Meltwater input estimates derived from Red Sea sea-level records (Fig. 4f, blue, fSL) on the revised chronology (section 5) display a large (up to 0.3 Sv) 1000-year-long meltwater pulse centred at 137 ka, a broad meltwater input between 134 and 131 ka with a large peak at 131.7 ka, and two relatively late meltwater pulses centred at 130 ka and 128.3 ka, respectively.



The magnitude and length of the AMOC perturbation resulting from such a meltwater scenario will, of course, depend on each model's sensitivity and on the initial AMOC state. However, for most models, it is expected that NADW formation would weaken significantly between ∼137.8 and 136.5 ka as well as between ∼133.5 and 129 ka. Another small AMOC perturbation is expected between ∼129 and 127.8 ka in this scenario.

There are significant uncertainties associated with both the simulation of NH ice-sheets and the timing and amplitude of sea-level changes. In addition, to fully explore the potential of transient simulations of TII, it is critical to simulate NADW changes in global agreement with proxy records. Since periods of increased IRD delivery have been associated with changes in NADW (e.g. van Kreveld et al., 2000; Rodrigues et al., 2017), we design an additional meltwater scenario, for which the timing is based on North Atlantic and Norwegian Sea IRD records (Fig. 8a). To construct the fIRD scenario, the IRD record of

MD95-2010 (Risebrobakken et al., 2007) is assumed to represent freshwater input only originating from the Eurasian ice-sheet for the period 140 to 133.8 ka, while the stack of IRD records presented in Figure 8b (black) is assumed to represent freshwater originating from both the North American and Eurasian ice-sheets for the period 133.7 to 127 ka. The normalized MD95-2010 IRD record and the IRD stack were thus scaled so as to obtain a total NH sle contribution of 87 m (Fig. 4e, red), with 35 m originating from the Eurasian ice-sheet and 51 m from the North American one. It is expected this scenario will lead to a

weakening of NADW between 139.5 and 136.5 ka as well as at ∼135 ka. The sustained meltwater pulse might induce NADW cessation between ∼133.5 and 129.4 ka, followed by a recovery sometime between 129 and 128 ka.

    As can be seen in Figure 4f, the meltwater forcing scenarios based on sea-level changes (fSL) and the IRD record (fIRD) share some similarities. The main differences between these scenarios are the small meltwater pulse at ∼137 ka in fSL, which is of much reduced amplitude in fIRD, and the ∼128 ka pulse in fSL not present in fIRD. Finally, the fSL scenario includes

periods of significant negative meltwater forcing (i.e. artificial salt flux addition), corresponding to phases of sea-level lowering. As described above, it is suggested to use the self-consistent drainage scheme for the location of the meltwater input. For NH scenarios fSL and fIRD, $(dH/dt)_{ice-sheet}$ (from equation 1) would need to be scaled to obtain the appropriate meltwater flux.

    For those who wish to take part in an inter-comparison involving comparable boundary conditions, the fSL scenario is put forward as the recommended option (Table 1). However, the ultimate choice of the appropriate freshwater scenario is left to

each group, and sensitivity experiments to assess the climatic impact of different meltwater scenarios are encouraged (section 7).

    To take into account the effect of Antarctic ice-sheet melting, freshwater should also be added close to the Antarctic coast, following the self-consistent routing scheme described above (Fig. 4d) (Briggs et al., 2014). This scenario will broadly consist in adding ∼0.0135 Sv meltwater from 140 to 130 ka. However, there are significant uncertainties associated with the timing

of the Antarctic deglaciation. Additional experiments are necessary to further constrain the impact of the Antarctic ice-sheet disintegration on the deglacial climate and carbon cycle (e.g. Menviel et al., 2010). Therefore, another scenario inspired by the Antarctic ice-sheet deglacial trajectory described in Goelzer et al. (2016) is proposed (Table 2, fSL2).

    Finally, if preferred and instead of the Northern and Southern Hemisphere freshwater scenarios described here, a globally uniform freshwater flux that corresponds to the ice-sheets evolution can be added to simply conserve salinity throughout the





transient deglacial experiment (Table 2, fUN). This latter option is equivalent to the *melt-uniform* scenario used in the PMIP4 transient simulations of the last deglaciation (Ivanovic et al., 2016).

## 7 Sensitivity experiments

The penultimate deglaciation is a particularly interesting period as it provides a framework to study the impact of changes in insolation, GHG concentrations and continental ice-sheets on climate. In addition, meltwater input associated with the disintegration of continental ice sheets will also impact the oceanic circulation and thus global climate and biogeochemistry (e.g. Liu et al., 2009; Menviel et al., 2011, 2014; Schmittner and Lund, 2015; Goelzer et al., 2016; Ivanovic et al., 2017; Menviel et al., 2018). Ultimately, transient simulations of TII will inform on the impact of each of these processes as well as their interactions.

However, the transient simulation of TII proposed here might present a challenge to state-of-the-art Earth-system models, as they will need to include a spin-up with 140-ka boundary conditions and be run for 13,000 years. For this reason (i.e. to avoid the necessity for additional simulations when the computational expense is prohibitive), the main experiment includes all appropriate boundary forcing as well as meltwater input due to disintegrating ice sheets. This experiment will provide valuable information on processes occurring during TII and will allow for a thorough comparison of the penultimate and the last deglaciation by complementing existing transient simulations of TI (Liu et al., 2009; Menviel et al., 2011; Roche et al., 2011; Gregoire et al., 2012; He et al., 2013) and the new experiments performed as part of PMIP4 (Ivanovic et al., 2016). These transient simulations of TII will also complement existing transient simulations of TII performed with the LOVECLIM Earth-system model and covering the time interval 135 to 115 ka (Loutre et al., 2014; Goelzer et al., 2016). Finally, the proposed experiment will provide a link to the PMIP4 transient simulation of the LIG (127 to 121 ka) as well as the PMIP4 127 ka timeslice experiment (*lig127k*) (Otto-Bliesner et al., 2017), even though the protocol of the LIG experiments includes pre-industrial continental ice-sheets.

The main experiment is described in Table 1 and includes a comprehensive meltwater scenario. Additional transient simulations of TII are encouraged to, for example, assess different timing and amplitude of meltwater-input (Table 2), but also simulations with globally uniform meltwater input (fUN). As there are large uncertainties associated with meltwater input scenarios and the sensitivity of deep convection to the freshwater input, it is strongly advised to perform both the main experiment and fUN to isolate the impact of the freshwater input (Goelzer et al., 2016). In addition, the response to individual forcings (i.e. orbital parameters, GHGs and ice-sheet extent and albedo) could be assessed separately (He et al., 2013; Gregoire et al., 2015).

Even though there are some geological constraints on glacial evolution (see Section 4.1.), there remain large uncertainties associated with the reconstruction of continental ice sheets during MIS 6, across TII and during MIS 5e. In addition, there are significant uncertainties associated with the parametrization of dynamical processes governing continental ice sheets, and most importantly those representing the climate forcing. Through their impact on albedo and topography, continental ice-sheets can significantly influence climate (e.g. Timm et al., 2010; Zhang et al., 2014), even when the ice-sheets are small (e.g.



Löfverström et al., 2014; Gong et al., 2015; Roberts and Valdes, 2017; Gregoire et al., 2018). Therefore, the impact of ice-sheet extent and topography on MIS 6, across TII and during MIS 5e, should be studied in detail through sensitivity experiments.

Models that include prognostic aerosol and/or dust could provide very useful fields to the community. Similarly, evaluating the impact of varying deglacial dust levels on climate and biogeochemistry are of crucial importance (Evan et al., 2009;
Martinez-Garcia et al., 2014). Therefore, sensitivity simulations forced with different dust-flux scenarios are encouraged.

## 8 Comparing model simulations to paleoclimate and paleoenvironmental reconstructions

It is central to PMIP to evaluate the realism of the coordinated transient simulations and the associated sensitivity experiments with environmental and climate reconstructions from archives. In the following, we first report on the few existing surface temperature syntheses that cover at least part of our interval of interest. We then provide a non-exhaustive selection of additional
paleoclimatic and paleoenvironmental records extending back to 140 ka. This will allow a first-order comparison between the changes recorded in different parts of the climate system and those inferred in the transient simulations (Figs. 8 and 9, Tables 3 and 4). The selection criteria are related to each record's temporal resolution and to the climatic or environmental interpretation that can be inferred from the measured tracer. We put a special emphasis on selecting marine sediment core records that can inform on millennial-scale changes occurring in the North Atlantic sector during HS11, potentially linked
with changes in NADW formation. This section ends with a review of the main limitations that will be associated with the model-data comparison and recommendations.

### 8.1 Available surface temperature syntheses

Quantitative comparisons between paleo-reconstructions and model outputs across the time interval 140-127 ka are possible, but remain limited to a few parameters (e.g. surface-air and sea-surface temperature (SAT and SST)). Qualitative and indirect
comparisons are also adequate to evaluate simulations: for example, simulated AMOC strength against paleo-records (section 8.2), simulated precipitation compared to Chinese speleothem calcite $\delta^{18}$O records or the simulated vegetation patterns against pollen-based vegetation reconstructions (section 8.3, Table 4).

There are currently no paleoclimate data compilations focusing on the penultimate glacial maximum (MIS 6) or covering the full length of TII, but syntheses of quantitative and temporal surface temperature changes focusing on the LIG are available
for model evaluation of the second part of the penultimate deglaciation. Indeed, the synthesis by Capron et al. (2014) covers the time interval 135-110 ka and compiles, in a coherent temporal framework underpinned by the AICC2012 ice-core chronology, annual SAT above Greenland and Antarctica and summer SST records from the North Atlantic and the Southern Ocean located at latitudes above 40°. Note that in Uemura et al. (2018), updated surface air temperature reconstructions displayed on AICC2012 are presented for the EDC, Dome F and Vostok deep ice cores. They represent useful constraints, particularly to
investigate spatial differences in temperature between East Antarctica inland ice core sites during TII.

Further, based on harmonized chronologies between marine records, Hoffman et al. (2017) propose a global annual SST compilation with timeseries encompassing 130 to 115 ka. Note that Hoffman et al. (2017) use the SpeleoAge age scale as a





reference chronology. This SpeleoAge age scale results from the adjustment of the EDC3 ice-core timescale using radiometric dates from Chinese speleothems (Barker et al., 2011) and presents age differences across TII with AICC2012 (e.g. ∼1.4 kyr at 127 ka, ∼1.0 kyr at 136 ka). Both syntheses are useful to perform site-to-site model-data comparisons in order to provide detailed information about the spatial structure of the changes. The regional stacks produced by Hoffman et al. (2017) provide

a first order estimate of the mean annual SST responses between 130 and 127 ka.

## 8.2 North Atlantic records to inform on Heinrich stadial 11 and NADW changes

Deglaciation of NH ice-sheets led to increased meltwater input into the North Atlantic, thus reducing surface water density and potentially weakening NADW formation. It has been shown that changes in NADW have a significant impact on North Atlantic and European climate (e.g. Stouffer et al., 2007; Martrat et al., 2014; Wilson et al., 2015), but also have a strong imprint on

tropical hydrology (e.g. Arz et al., 1998; Chiang and Bitz, 2005; Deplazes et al., 2013; Otto-Bliesner et al., 2014; Jacobel et al., 2016). Through oceanic and atmospheric teleconnections, NADW variations can further modulate the strength of the Asian Monsoon (e.g. Cheng et al., 2009) and the formation of other oceanic water masses, such as North Pacific Intermediate Water (Okazaki et al., 2010). To maximise the use of transient simulations of TII we investigate here, using a multi-proxy approach, the millennial-scale variability occurring during TII, and particularly potential changes in NADW linked with the meltwater

input of HS11.

We selected 11 marine sediment cores from the Western Mediterranean Sea, the North Atlantic and the Nordic Seas with relatively high-resolution proxy data across TII (Tables 3 and 4, Fig. 8). These sites have undergone multi-proxy investigations (e.g. stable isotopes on benthic and planktonic foraminifera, foraminifera faunal assemblages, IRD counts, SST reconstructions, Pa/Th, $\epsilon_{Nd}$). The chronology of all sites has been revised here, using the radiometrically-dated time scale of the Italian Corchia

cave (Drysdale et al., 2009, 2018) as a reference for TII (Text S1 and S2, Table S1).

Phases of meltwater input, and/or changes in NADW can be investigated using different proxies. The intensity of NADW formation can be deduced from variations in Pa/Th (a kinematic proxy for the rate of deep water export from the North Atlantic) and $\epsilon_{Nd}$ (a water provenance tracer) in North Atlantic sediment cores (McManus et al., 2004; Roberts et al., 2009), but only one record (ODP 1063) currently covers TII and the time-resolution of Pa/Th and $\epsilon_{Nd}$ measurements across TII is relatively low

(Böhm et al., 2015; Deaney et al., 2017). Phases of iceberg discharges, which might significantly impact NADW formation, can be inferred from the amount of IRD in North Atlantic cores. For this time period, IRD records with sufficient resolution are available from six Atlantic cores (CH69-K09, ODP980, ODP983, ODP984, SU90-03 and ODP1063) and one Norwegian Sea core (MD95-2010). The IRD records were first normalized, after which an IRD stack was made for the North Atlantic cores by interpolating them onto a common age-scale with a regular (100-year) time-step.

Because continental ice-sheets are $^{18}$O-depleted, the meltwater supply also induces a drop in seawater $\delta^{18}$O ($\delta^{18}$Osw), particularly at the ocean's surface. For six of the sites, changes in $\delta^{18}$Osw have been reconstructed from paired SST and planktic $\delta^{18}$Oc records using the following paleo-temperature equation (Shackleton, 1974):

$$Tiso = 16.9 - 4.38 * (\delta^{18}Oc + 0.27 - \delta^{18}Osw) + 0.10 * (\delta^{18}Oc + 0.27 - \delta^{18}Osw)^2 \qquad (2)$$



with *Tiso* the isotopic or calcification temperature (°C) that is deduced from SST reconstructions; $\delta^{18}$Oc, the isotopic composition of the calcite (‰, PDB); $\delta^{18}$Osw, the isotopic composition of seawater (‰, SMOW). The factor 0.27 is added for calibration against international standards (SMOW vs. VPDB) (Hut, 1987).

As a weakened NADW transport leads to the accumulation of remineralized carbon below 2500 m in the North Atlantic (Marchal et al., 1998), changes in oceanic circulation are also inferred from benthic foraminifera $\delta^{13}$C, a tracer for the ventilation of deep-water masses (Duplessy et al., 1988; Curry et al., 1988; Menviel et al., 2015; Schmittner et al., 2017). Finally, since a NADW weakening reduces the meridional heat transport to the North Atlantic (Stouffer et al., 2007), it is generally accompanied by a sea surface cooling across this region (Kageyama et al., 2013; Ritz et al., 2013).

The Norwegian Sea core MD95-2010 (Risebrobakken et al., 2006) indicates an increased IRD content starting at 139.5 ka
(Fig. 8). This could suggest an initiation of the penultimate deglaciation by enhanced iceberg calving (and subsequent melting) from the Fennoscandian ice-sheet. However, this has little to no effect in the North Atlantic. Only in the western side of the North Atlantic, right in the meanders of the Gulf Stream, does core CH69-K09 display a surface cooling and a drop in $\delta^{18}$Osw starting at ∼137.5 ka (Labeyrie et al., 1999).

A more significant deglacial pulse, indicated by an IRD increase in both the Nordic Seas and the North Atlantic, as well
as a surface freshening in all North Atlantic cores occurs at ∼136.4-133.9 ka (Fig. 8g). This meltwater input, which could correspond to the first phase of HS11, could have led to a significant weakening of NADW, as indicated by $\epsilon_{Nd}$ and Pa/Th records from the Bermuda rise (Böhm et al., 2015) (not shown), as well as by the accompanying surface cooling in the North Atlantic (Chapman and Shackleton, 1998; Martrat et al., 2007, 2014; Deaney et al., 2017) and a decrease in benthic $\delta^{13}$C below 2000 m depth (Labeyrie et al., 1999; Shackleton et al., 2003; Hodell et al., 2008; Deaney et al., 2017). The low-resolution $\epsilon_{Nd}$
and Pa/Th records from the Bermuda rise (Böhm et al., 2015) also indicate a weakening of NADW transport during this period (not shown). This deglacial phase identified in North Atlantic marine sediment cores is broadly coincident with MWP-2A (Fig. 7, section 5).

Martrat et al. (2014) identified a double-u structure in SST records from the western Mediterranean Sea (ODP976) and the Portuguese margin (MD95-2042, MD01-2444) during TII, with a short-duration warming centred at ∼133.6 ka. This warming
is also seen in other North Atlantic records, such as ODP 984, 983, 980 and 1063 (Fig. 8 c-f). Very interestingly, and within dating uncertainties, this event corresponds to a small increase in atmospheric $CH_4$ (Figs. 1 and 2) (Landais et al., 2013), and broadly corresponds to a pause in the sea-level rise. In parallel, Pa/Th in core ODP1063 displays a significant decrease at this time (not shown), and benthic $\delta^{13}$C increases in the deepest North Atlantic cores (Fig. 8). This warming could thus be due to a short lived reinvigoration of NADW formation.

In agreement with previous studies (e.g. Marino et al., 2015), the main phase of HS11 occurs between ∼133.3 and 130.4 ka and is characterised by a large IRD peak, cold surface-water conditions, a minimum in seawater $\delta^{18}$O , and low benthic $\delta^{13}$C values in most sites of the North Atlantic (Fig. 8). Elevated rates of iceberg calving and melting of NH ice sheets thus lead to a large meltwater pulse in the North Atlantic, a significant global seal level rise (MWP2-B) (Grant et al., 2012, 2014), NADW weakening (Böhm et al., 2015) and a surface cooling that is at least regional.





The resumption of NADW formation at the end of HS11 marks the end of the penultimate deglaciation (Tzedakis et al., 2012) ($\sim$130.4-128.5 ka). It is characterised by an increase in SST, seawater $\delta^{18}$O (Oppo et al., 2006; Mokeddem and McManus, 2016; Martrat et al., 2007, 2014; Deaney et al., 2017) and benthic $\delta^{13}$C from the North Atlantic region (Fig. 8). The associated atmospheric warming in the North Atlantic region also leads to terrestrial regrowth and thus a sharp increase in atmospheric

CH$_4$ occurring between $\sim$129 and 128.5 ka (Landais et al., 2013).

## 8.3   Other environmental and climate reconstructions

Figure 9 shows a non-exhaustive selection of other environmental and climate proxy records. Tables 3 and 4 further provide the climate or environmental interpretation for each selected tracer to aid the model-data comparison, including the tracers described in section 8.2. Only a few speleothem calcite $\delta^{18}$O ($\delta^{18}$Oc) records and their environmental or climatic interpretation

are presented here. However, it can be complemented with the compilation of speleothem $\delta^{18}$Oc records covering the time interval 140 to 110 ka presented by Govin et al. (2015). It is also important to recognise that the dominant drivers of speleothem $\delta^{18}$Oc may change over time and differ from one cave site to another (see Section 8.4). Therefore we strongly advise a critical assessment of these interpretations based on the most recent developments and advances in stable-isotope hydrology and the original publications.

Tables 3 and 4 also detail the timing of the major changes (and associated uncertainties) as recorded in each timeseries. These major changes are identified using the RAMPFIT or BREAKFIT software programs (Mudelsee, 2000, 2009). Age uncertainties ($1\sigma$) reported in Tables 3 and 4 include *i)* the "internal" error of the event given by RAMPFIT or BREAKFIT, *ii)* the relative error related to the climatic alignment method for marine sediment and Ioannina records and *iii)* the absolute dating error of the reference time scale. Note that the selected ice- and sediment-based records are all displayed on AICC2012 or on a timescale

coherent to this ice-core age scale, while terrestrial records are displayed on their own independent chronology. Among these paleoclimatic time series, several isotopic records are presented here and we strongly encourage transient simulations to be performed with oxygen and/or carbon isotope-enabled models to allow direct quantitative comparison between simulated and measured isotopic time series.

## 8.4   Limitations and recommendations

One important consideration to account for when comparing simulated variables against paleodata between 140 and 127 ka, is the large uncertainties associated with both absolute and relative chronologies of most paleoclimatic records during this time interval. Dating uncertainties range from a few centuries to up to several thousand years depending on the type of archive and dating methods (Govin et al., 2015). For instance, the average absolute dating error attached to the Corchia Cave records is $\sim$0.7 kyr ($2\sigma$) (Drysdale et al., 2018). For marine sediment chronologies, which are mainly based on record alignment

strategies (e.g. a record on a depth scale is aligned onto a dated reference record), the age uncertainties encompass a relative dating error ("alignment error") in addition to the absolute error attached to the chronology of the dated reference record. As a result, the overall $2\sigma$ age error associated with North Atlantic sediment core ODP976 aligned onto the Corchia record is 1.6 kyr on average (Table S2). Between 140 and 127 ka, the average absolute error attached to the AICC2012 chronology




used to display ice and gas records from the EDC ice core is about 4 kyr ($2\sigma$) (Bazin et al., 2013; Veres et al., 2013). This large AICC2012 dating uncertainty is thus attached to the GHG concentration records used to force the transient simulations (Section 3). It will therefore taint the relative timing of the changes in orbital and atmospheric $CO_2$ forcing that will largely drive the simulated evolution of climate and environmental changes across TII. The relative timing of those changes will also be affected

by uncertainties attached to the temporal evolution of the ice sheets and related meltwater forcing scenarios. Note, however, that uncertainties attached to the relative timing of changes between GHG forcing and the simulated changes in Antarctica and the Atlantic Ocean basin are somewhat reduced since the ice- and sediment-based records are also displayed on AICC2012 timescale or on time scales coherent to the reference ice-core age scale (Section 8.2).

Limitations are also attached to the potential misinterpretation of climate and environmental proxies due to incomplete
understanding of how some of those archives record climatic and environmental change. First, SST records highlighted here have been reconstructed using various microfossil and geochemical methods. Although the use of various tracers is known to yield SST discrepancies in particular above 35°N (MARGO project members, 2009), the extent to which these different SST reconstruction methods influence the representation of temporal climatic changes across our studied time interval is poorly known. Additional difficulties arise from the individual methods commonly used to reconstruct past SST due to, for example,
the poor understanding of the modern habitat (e.g. living season and water-depth) of microfossil species (e.g. foraminifera) (Jonkers and Kučera, 2015) and alkenone producers (e.g. Rosell-Melé and Prahl, 2013). With these limitations acknowledged, model-data comparisons should use annual or appropriate seasonal climatic variables depending on the interpretation of the measured climate proxy proposed in the original or subsequent publications.

Regarding SAT reconstructions over Antarctica, the impact of changes in seasonality of snow precipitation on the reconstruc-
tion remains difficult to quantify based on ice-core water-isotopic records (e.g. Masson-Delmotte et al., 2011; Uemura et al., 2012). In addition, Sime et al. (2009) suggested that the temperature at Dome C should be much higher than the one inferred from water isotopes and a constant temperature/$\delta^{18}$O slope during MIS 5e. For now, we suggest the use of simulated annual mean climatic variables for the comparison. However, it is crucial that the seasonality of paleo-records is better assessed to improve the interpretation of temperature reconstructions, and hence the model-data comparisons.

In addition, uncertainties remain regarding the dominant controlling factors (i.e. changes in temperature, rainfall amount and rain sources) of speleothem calcite isotopic records (e.g. Govin et al., 2015). For instance, $\delta^{18}$Oc records throughout Asia are commonly interpreted as tracers of past changes in the intensity of the Asian monsoon. In particular, Chinese speleothem $\delta^{18}$Oc is classically interpreted as reflecting the East Asia monsoon (e.g. Cheng et al., 2009, 2016). However, a water-hosing experiment performed with an oxygen-isotopes-enabled model suggests instead that $\delta^{18}$Oc variations may reflect changes in
the intensity of the Indian, rather than East Asian, monsoon precipitation during Heinrich events (Pausata et al., 2011). Another recent model study also demonstrates that variations in ice core and speleothem oxygen isotope reconstructions cannot solely be attributed to climatic effects, but also reflect depleted $\delta^{18}$O of nearby oceans during glacial meltwater events (Zhu et al., 2017). Overall, we strongly encourage the use of isotope-enabled models to allow for direct and quantitative model-data comparison of isotopic tracers.



Additional multi-centennial-scale paleoclimatic reconstructions are necessary in order to further constrain the millennial scale variability during TII. It is also crucial that comprehensive data compilation work is carried out over the entire studied time interval and covering, in particular, the penultimate glacial maximum in order to test the robustness of the initial 140 ka spin-up climate. Modelling groups running transient simulations of TII, and/or associated sensitivity experiments, are encouraged to
use multiple paleorecords for a full diagnosis of the simulations.

## 9    Conclusions

Here, we present a protocol for performing transient simulations of TII spanning 140 to 127 ka. Changes in boundary conditions across TII that will serve as a forcing are presented and discussed. This includes changes in orbital parameters, GHG concentration, NH and Antarctic ice sheets, and associated deglacial meltwater input. While not used as a direct forcing,
changes in global sea-level are also presented on a new chronology. Finally, a series of key paleoclimatic and paleoenvironmental records are suggested to perform model-data comparisons. The marine records were recovered from the North Atlantic and Southern Ocean, while the continental records were retrieved from Europe, China and Antarctica. Performing transient simulations with oxygen- and/or carbon-isotope-enabled Earth-system models could significantly improve model-data comparisons by providing a more direct and quantitative comparison with paleoproxies based on measured isotopic signatures (e.g.
$\delta^{18}$O, $\delta^{13}$C).

Simulations of the penultimate deglaciation would allow a comparison with the last deglaciation, therefore highlighting similarities and differences between the last two deglaciations. The evolution of insolation across the two terminations is different, potentially explaining the relatively more rapid disintegration of NH ice-sheets during TII than TI. Acting both as a response to and driver of changes, atmospheric $CO_2$ appears to increase much more gradually during TII than TI. Another
striking difference between TII and TI is the lack of a Bølling-Allerød warming in the middle of the deglaciation, even though, as discussed in sections 5 and 8, several records show a brief reversal centred at about 133.6 ka. Transient simulations can thus shed light onto the different and interactive roles of radiative forcing (insolation, GHG concentrations) and ocean circulation changes (e.g. NADW and Southern Ocean ventilation) in driving climate changes across TII and TI. Transient simulations performed with Earth-system models that include a global carbon cycle model, would be particularly useful in assessing the
processes leading to the different atmospheric $CO_2$ increase trajectories across the two deglaciations.

Transient simulations of TII and associated model/paleo-proxy comparisons provide a great opportunity to understand drivers and processes involved in past climate change. In addition, these transient simulations of TII will provide a bridge with the proposed PMIP4 transient simulations of the LIG (127 to 121 ka), and the 127-ka time-slice experiment (*lig127k*) (Otto-Bliesner et al., 2017). TII is a particularly interesting period to understand as it led to the LIG, an interglacial displaying
warmer conditions than pre-industrial (e.g. Hoffman et al., 2017) as well as a global sea-level 6 to 9 m higher than today (e.g. Dutton et al., 2015).





*Data availability.* All the forcing files as well as the paleo-data described in the manuscript will be available on the PMIP4 website
https://pmip4.lsce.ipsl.fr/doku.php/exp_design:index upon publication.

*Acknowledgements.* This study was performed as part of the PAGES-PMIP working group on Quaternary Interglacials (QUIGS) and was
initiated during the second QUIGS workshop at the Université du Quebec in Montreal, 18-20 October 2016. LM acknowledges funding

5   from the Australian Research Council grants DE150100107 and DP180100048. EC received funding from the European Union's Seventh
Framework Programme for research and innovation under the Marie Skłodowska-Curie grant agreement nb 600207. EW is supported by a
Royal Society professorship. His part in this project has received funding from the European Research Council (ERC) under the European
Union's Horizon 2020 research and innovation programme (grant agreement nb 742224). PCT and RFI acknowledge funding from from
NERC (NE/G00756X/1 to PCT and NE/K008536/1 to RFI). AD acknowledges U.S. National Science Foundation (NSF) grants 1559040

10  and 1702740, as well as the PALSEA working group. BOB is supported by the National Center for Atmospheric Research (NCAR), which
is sponsored by the NSF. Her part in this project received funding from NSF P2C2 award 1401803. F.H. was supported by the NSF (AGS-
1502990) and by the NOAA Climate and Global Change Postdoctoral Fellowship program, administered by the University Corporation for
Atmospheric Research. XZ is supported by Helmholtz Postdoc Program (PD-301) and the Chinese "The Thousand Talents Plan" Program.
AA, KK and IO acknowledge support by JSPS KAKENHI grants (17H06104 to AA, 15KK0027 and 26241011 to KK, 17K12816 and

15  17J00769 to IO), and MEXT KAKENHI grant (17H06323 to AA and 17H06320 to KK).



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



| Forcing | 140 ka spin up | Transient simulation (140-127 ka) |
|---|---|---|
| **Orbital parameters** | | |
| Eccentricity | 0.033 | from Berger (1978) |
| Obliquity (°) | 23.42 | from Berger (1978) |
| Perihelion - 180° | 251 | from Berger (1978) |
| **Atmospheric greenhouse gases concentrations** | | |
| on the AICC2012 chronology (Bazin et al., 2013; Veres et al., 2013) | | |
| $CO_2$ | 195 ppm | EDC record from Bereiter et al. (2015) |
| $CH_4$ | 387 ppb | EDC record from Loulergue et al. (2008) |
| $N_2O$ | 201 ppb | 201 ppb from 140 to 134.5 ka |
| | | followed by EDC record from Flückiger et al. (2002) |
| **Ice-sheets** | | |
| North American and Eurasian | 140 ka | IcIES-NH (Abe-Ouchi et al., 2013) |
| Greenland | 140 ka GSM-G | GSM-G (Tarasov et al., 2012) |
| Antarctica | 140 ka GSM-A | GSM-A (Briggs et al., 2014) |
| **Bathymetry and orography** | | |
| Bering Strait | closed | gradual opening consistent with sea-level rise |
| Sunda and Sahul shelves | emerged | gradual flooding consistent with sea-level rise |
| **Freshwater input** | | |
| Northern Hemisphere | none | based on sea-level changes (fSL, blue in Fig. 4f) |
| Antarctic coast | none | 0.0135 Sv between 140 and 130 ka (constant rate) |

**Table 1.** Summary of forcings and boundary conditions to apply for the penultimate glacial spin up (140 ka) and subsequent transient simulation of TII.





| Scenario | Northern Hemisphere | Antarctic coast | Globally uniform |
|---|---|---|---|
| fSL | based on sea-level changes (blue in Fig. 4f) | 0.0135 Sv - 140-130 ka | - |
| fIRD | based on IRD (red in Fig. 4f) | 0.0135 Sv - 140-130 ka | - |
| fIC | based on ice-sheet changes (black in Fig. 4f) | 0.0135 Sv - 140-130 ka | - |
| fSL2 | based on sea-level changes (blue in Fig. 4f) | triangular input with | - |
| | | max 0.15 Sv - 131-128 ka | - |
| fUN | - | - | based on sea-level changes |

**Table 2.** Freshwater scenarios for transient simulations of TII. Freshwater can be added in the NH and close to the Antarctic coast based on meltwater routing, or as a globally uniform flux. The Antarctic freshwater forcing in fSL2 is based on Goelzer et al. (2016), but scaled down to obtain a total Antarctic contribution of 20 m sle.





| Tracer interpretation | Core | Coordinates and depth (m) | φ1 (ka) | φ2 (ka) | φ3 (ka) | φ4 (ka) | φ5 (ka) | References |
|---|---|---|---|---|---|---|---|---|
| **Sea-level** | | | | | | | | |
| Sea-level | **Red Sea cores** | - | 137.0±0.7 *increases* | | 133.4±0.7 *main increase* | | 130.2±1 | Grant et al. (2012) This study |
| **Benthic δ¹³C** | | | | | | | | |
| North Atlantic intermediate-depth | **ODP983** | 60.40°N, 23.64°W 1984 m | 136.1±1.2 *weaker ventil.* | | | | 128.1±0.9 | Raymo et al. (2004) Barker et al. (2015) |
| ventilation | **ODP980** | 55.80°N, 14.11°W 2180 m | 137±1.9 | | | 128.6±1.8 | 127.6±1.3 | Oppo et al. (2006) |
| North Atlantic deep-water | **MD95-2042** | 37.80°N, 10.17°E 3146 m | | | | 131.0±1.4 *stronger ventil.* (T) | | Shackleton et al. (2003) |
| ventilation | **Stack of U1308 CH69-K09 and ODP 1063** | 49.88°N, 24.24°W, 3883 m 41.76°N, 47.35°W, 4100 m 33.69°N, 57.62°W, 4584 m | 135.9±2.0 | | | 130.3±1.6 | 129.2±1.4 | Hodell et al. (2008) Labeyrie et al. (1999) Deaney et al. (2017) |
| Southern Ocean deep-water ventilation | **MD02-2488** | 46.48°S, 88.02°E 3420 m | | | 131.9±2.1 *stronger ventil.* (U) | 130.2±2.2 *weaker ventil.* (V) | | Govin et al. (2009) Govin et al. (2012) |
| **Planktic δ¹⁸O and δ¹⁸Osw** | | | | | | | | |
| North Atlantic surface δ¹⁸O | **ODP980** | 55.80°N, 14.11°W 2180 m | | | | 130.0±1.3 | | Oppo et al. (2006) |
| *salinity* | **SU90-03** | 40.51°N, 32.05°W | | | | 131.0±1.1 | | CS98 |
| | **MD95-2042 ODP 976** | 37.80°N, 10.17°E 36.20°N, 4.31°E | | | 133.9±0.9 | 131.6±1.5 131.9±0.9 | | Shackleton et al. (2003) Martrat et al. (2014) |
| **Speleothem δ¹⁸Oc** | | | | | | | | |
| North Atlantic surface δ¹⁸O | **Corchia Cave, Italy** | 43.97°N, 13.0°E, 840 m a.s.l | | | 133.9±1.2 *NA meltwater input* (I) | 131.0±0.7 *NA meltwater paused* (J) | | Drysdale et al. (2009) Drysdale et al. (2018) Marino et al. (2015) |
| **Mean ages for the beginning of φ1-φ5 from Tables 3 and 4** | | | | | | | | |
| Mean ages | | | 136.4±1.7 | 133.9±0.8 | 133.3±1.1 | 130.4±1.3 | 128.5±1.3 | |

**Table 3.** Key paleoproxy records of changes in ocean oxygen ($\delta^{18}$O) and carbon ($\delta^{13}$C) isotopic composition. The chronology of all paleoproxy records presented here is based on an alignment onto Corchia U-Th-based chronology (Table S1). Letters in brackets indicate the major changes identified in the paleoclimatic records shown in Figure 9. The dates of the major changes obtained through RAMPFIT or BREAKFIT are indicated with their associated uncertainties. Age uncertainties ($1\sigma$) reported include (1) the "internal" error of the event given by RAMPFIT (for most major changes) (Mudelsee, 2000), or BREAKFIT (major change dates highlighted with **) (Mudelsee, 2009), (2) the relative error related to the climatic alignment method for marine sediment records and (3) the absolute dating error of the reference time scale. Based on North Atlantic records, changes have been split into five phases ($\varphi$), with the date representing the beginning of each phase: $\varphi$1 is associated with the early phase of HS11 and MWP-2A, $\varphi$2 represents a pause within HS11, $\varphi$3 the main phase of HS11 and MWP-2B, $\varphi$4 the inception out of HS11 and $\varphi$5 full interglacial conditions. Implicitly the end of each phase corresponds to the beginning of the next one. For the sea-level record, the new chronology discussed in section 5 is used. CS98 refers to Chapman and Shackleton (1998). *ventil.* stands for *ventilation*.





| Tracer interpretation | Core, coordinates and depth/elevation | Chronology | $\varphi 1$ (ka) | $\varphi 2$ (ka) | $\varphi 3$ (ka) | $\varphi 4$ (ka) | $\varphi 5$ (ka) | References |
|---|---|---|---|---|---|---|---|---|
| **Atm. CO$_2$ concentration** | | | | | | | | |
| Atm. CO$_2$ concentration | **EDC** 75.05°S, 123.19°E, 3233 m asl | Ice core AICC2012 chrono. | 137.8±2.7 *increases* | | | | 128.0±1.8 | Bereiter et al. (2015) |
| **SST** | | | | | | | | |
| North Atlantic *Summer* SST (FFA) | **ODP980** 55.80°N, 14.11°W, 2180 m depth | Alignment onto Corchia U-Th-based chrono. | | | | 129.7±1.3 | 128.7±1.3 | Oppo et al. (2006) |
| North Altantic *Summer* SST (FFA) | **SU90-03** 40.51°N, 32.05°W, 2475 m depth | Alignment onto Corchia U-Th-based chrono. | 136.9±1.6 *colder* (E) | | | 131±1 *warmer* (F) | | CS98 Cortijo et al. (1999) |
| W. Mediterr. Sea SST (Uk'37) | **ODP 976** 36.20°N, 4.31°E 1108 m depth | Alignment onto Corchia U-Th-based chrono. | 135.9±1.1 *colder* (A) | 134 ±1.1 (B) | 133.3** ±0.9 (C) | 131.4±0.9 *warmer* (D) | | Martrat et al. (2014) |
| Southern Ocean *Summer* SST (FFA) | **MD02-2488** 46.48°S, 88.02°E, 3420 m depth | Alignment onto AICC2012 | | | | 130±2.1** *max. warm* (R) | | Govin et al. (2009) Govin et al. (2012) |
| Southern Ocean SST (Mg/Ca) | **MD97-2120** 45.54°S, 174.92°E, 1210 m depth | Alignment onto AICC2012 | | | | 128.1±2.5** *max. warm* (S) | | Pahnke et al. (2003) |
| **Air temperature (SAT)** | | | | | | | | |
| Antarctic *Annual* SAT (Ice δD) | **EDC ice core** 75.1°S, 123.35°E, 3233 m a.s.l. | Ice core AICC2012 chrono. | 135.6±2.5 *(warmer)* (X) | | | 129.4±1.8** *max. warm* (Y) | | Jouzel et al. (2007) |
| *SAT* Southern Europe ([Mg]) | **Corchia Cave, Italy** 43.97°N, 13.0°E, 840 m a.s.l | Alignment onto Corchia U-Th-based chrono. | | | | 131.0±0.7 *warmer* (K) | 128.6±0.6 *warm plateau* (L) | (Drysdale et al., 2018) |
| **Precipitation** | | | | | | | | |
| Vegetation/precipitation in Southern Europe | **Lago di Monticchio, Italy** 40.93°N, 15.58°E, 656 m a.s.l. | Independent absolute varve chronology | | | | | 127.2±1.6 *max. warm/wet* (G) | Brauer et al. (2007) |
| (Temperate tree pollen) | **Ioannina terr. sequence, Greece** 39.65°N, 20.91°E, 470 m a.s.l. | Orbital tuning | | | | 132.3±2.3 *wetter* (H) | | Tzedakis et al. (2003) |
| Intensity of Asian monsoon ($\delta^{18}$Oc) | **Chinese Caves** 25.28°N to 32.5°N 108.08-119.16°E 680 - 1900 m a.s.l. | Absolute U-Th-based chronology | 135.6±0.5 *drier* (M) | 133.7 ±0.5 (N) | 133.3** ±0.5 (O) | 128.9±0.1 *wetter* (P) | | Cheng et al. (2016) |
| Intensity of East Asian monsoon (grain size) | **Chinese loess** 35.62-37.14°N 103.20 - 109.85°E | Alignment onto Hulu-Sanbao U-Th-based chrono. | | | | 130.2±0.4 ∘ *wetter* (Q) | | Yang and Ding (2014) |
| **Mean ages for the beginning of $\varphi 1$-$\varphi 5$ from Tables 3 and 4** | | | | | | | | |
| Mean ages | | | 136.4±1.7 | 133.9±0.8 | 133.3±1.1 | 130.4±1.3 | 128.5±1.3 | |

**Table 4.** Same as Table 3 for key paleoproxy records of climatic and environmental changes through TII selected for their relatively high resolution. An indication of their climatic or environmental use is indicated in italic and the proxy used is shown in parenthesis in column 1. Age uncertainties (1σ) reported include (1) the "internal" error of the event given by RAMPFIT (for most major changes) (Mudelsee, 2000), or BREAKFIT (major change dates highlighted with **) (Mudelsee, 2009), (2) the relative error related to the climatic alignment method for marine sediment and Ioannina records and (3) the absolute dating error of the reference time scale. ∘ No dating error is provided in Yang and Ding (2014). As a result, the stated error here only encompasses the "internal" error, it thus represents only a minimal error for the timing of the stacked grain size increase. The mean age representing the beginning of each phase ($\varphi$) is shown in the last row and is calculated from all the available dates within each phase shown in Tables 3 and 4. FFA stands for foraminifera faunal assemblages. CS98 refers to Chapman and Shackleton (1998). Based on North Atlantic records, changes have been split into five phases ($\varphi$), with the date representing the beginning of each phase: $\varphi 1$ is associated with the early phase of HS11 and MWP-2A, $\varphi 2$ represents a pause within HS11, $\varphi 3$ the main phase of HS11 and MWP-2B, $\varphi 4$ the inception out of HS11 and $\varphi 5$ full interglacial conditions. Implicitely the end of each phase corresponds to the beginning of the next one.



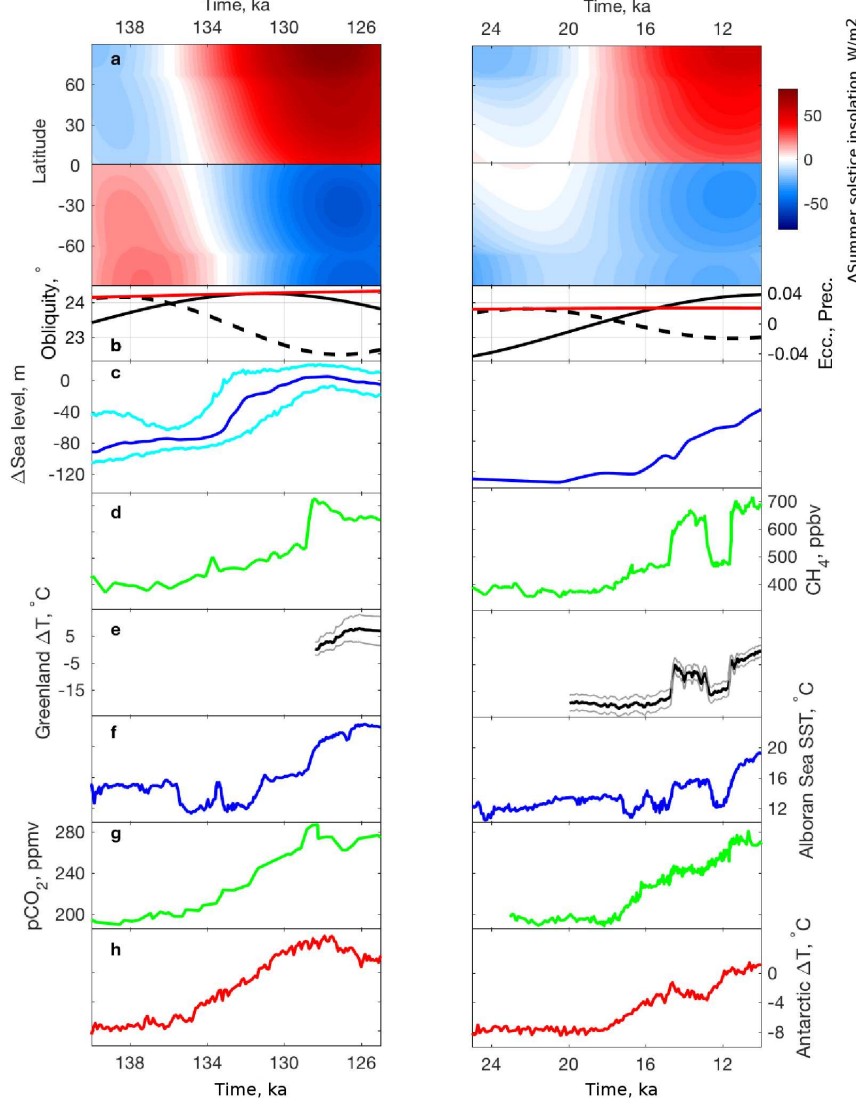

**Figure 1. Overview of (left) TII and (right) TI climatic and environmental evolutions: a)** Hovmöller diagram of summer solstice insolation anomalies (W/m$^2$); Timeseries of **b)** eccentricity (red), obliquity (solid black) and precession (dashed black) (Berger, 1978); **c)** Global mean sea level anomaly probability maximum (m, blue), (left) including its 95% confidence interval (cyan) (Grant et al., 2014), from Grant et al. (2012) on the original age model for TII and (right) from Lambeck et al. (2014) for TI. **d)** Atmospheric methane (CH$_4$) concentration as recorded in EDC ice core, Antarctica (Loulergue et al., 2008); **e)** Precipitation-weighted surface temperature temperature reconstruction based on stable water isotopes from the Greenland NEEM ice core (left), and annual surface temperature composite reconstruction based on air nitrogen isotopes from the Greenland NEEM, NGRIP and GISP2 ice cores (Buizert et al., 2014); **f)** Alkenone-based (Uk'$_{37}$) SST reconstruction from ODP976 (Martrat et al., 2014); **g)** Atmospheric CO$_2$ concentration as recorded in Antarctic ice cores (left) EDC (Bereiter et al., 2015) and (right) WAIS divide (Marcott et al., 2014) on the WD2014 chronology (Buizert et al., 2015); **h)** Antarctic temperature anomalies relative to present day inferred from EDC ice core (Jouzel et al., 2007). Unless specified, all records are displayed on the AICC2012 timescale (Bazin et al., 2013; Veres et al., 2013) or a chronology coherent with AICC2012.





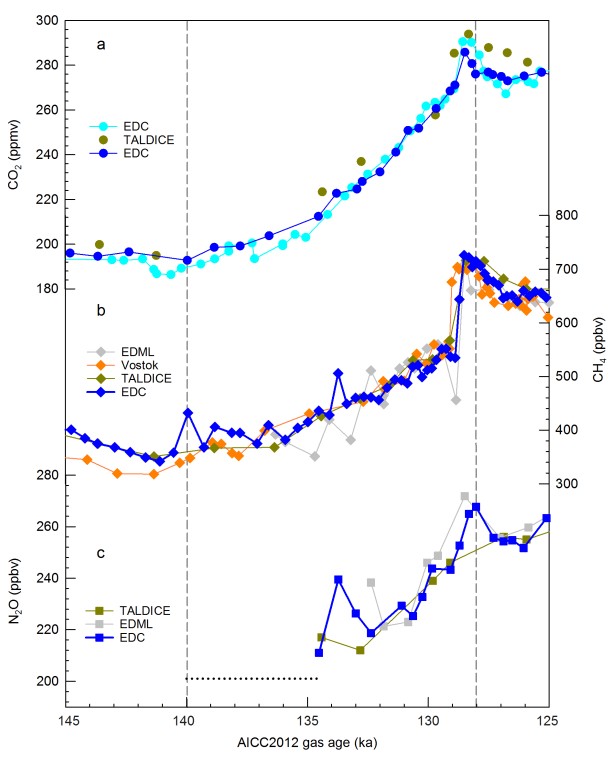

**Figure 2. Atmospheric greenhouse gas concentrations:** Atmospheric trace gases through the penultimate deglaciation from Antarctic ice cores displayed on the AICC2012 chronology (Bazin et al., 2013; Veres et al., 2013). **a)** Atmospheric $CO_2$ concentrations from EDC (turquoise and blue) (Lourantou et al., 2010; Schneider et al., 2013) and TALDICE (green) (Schneider et al., 2013). **b)** Atmospheric $CH_4$ concentration from EDC (Loulergue et al., 2008) (blue), Vostok (Petit et al., 1999) (orange), TALDICE (Buiron et al., 2011) (green) and EDML (Capron et al., 2010) (grey). **c)** Atmospheric $N_2O$ concentration from EDC (Flückiger et al., 2002) (blue), EDML (Schilt et al., 2010) (grey) and TALDICE (Schilt et al., 2010) (green). Due to in-situ production within the ice sheet, no accurate $N_2O$ measurements are available beyond 134.5 ka. Between 140 and 134.5 ka a constant $N_2O$ level of 201 ppb should be used (dashed black line).





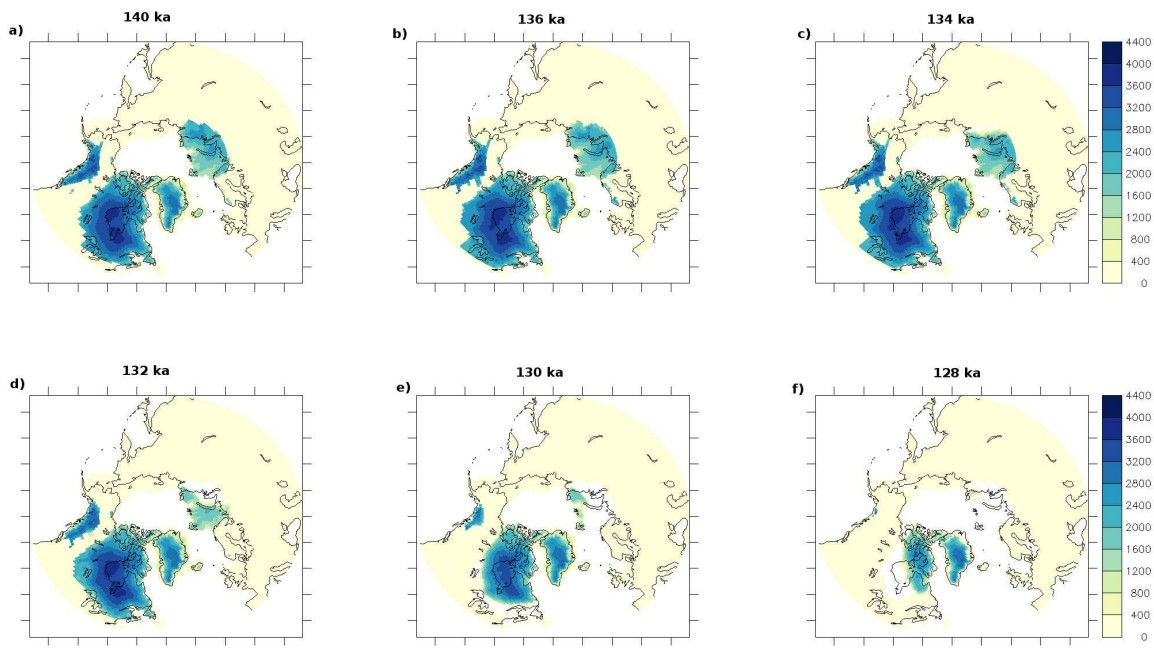

**Figure 3. NH ice-sheets** elevation (m) at **a)** 140, **b)** 136, **c)** 134, **d)** 132, **e)** 130 and **f)** 128 ka from the combined ice-sheet forcing: as simulated by the IcIES-MIROC model (Abe-Ouchi et al., 2013) for the North American and Eurasian ice-sheets, and as simulated by the Glacial Systems Model (GSM) (e.g. Tarasov et al., 2012) for the Greenland ice-sheet. Elevation is shown where the ice-mask is greater than 0.5.



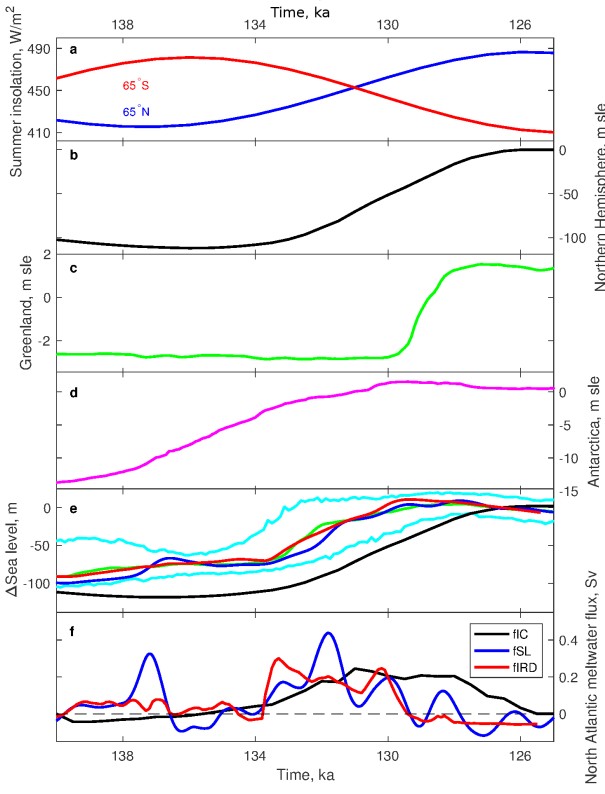

**Figure 4.** Timeseries of **a)** Summer solstice insolation at 65°N and 65°S, **b-d)** global sea level equivalent (m) of changes in **b)** NH (Abe-Ouchi et al., 2013), **c)** Greenland (Tarasov et al., 2012) and **d)** Antarctic ice-sheets (Briggs et al., 2014), **e)** global sea level (m) change estimated from Red Sea records (Grant et al., 2014) (green) with the 95% probability interval (cyan), sea-level record with an adjusted age scale as described in section 5 (blue), and as estimated from the continental ice-sheets simulations (black). The red line shows the changes in global sea-level that would be obtained by adding the meltwater flux, fIRD, described in **f** plus an Antarctic contribution of 13 m. **f)** Possible North Atlantic meltwater flux (Sv) scenarios: estimated from the disintegration of NH ice-sheets as shown in **b, c** (Tarasov et al., 2012; Abe-Ouchi et al., 2013) (black), estimated from the global sea-level change on the revised age-scale (this study, blue), and scaled from the North Atlantic and Norwegian Sea IRD records (red, shown in Fig. 8b).





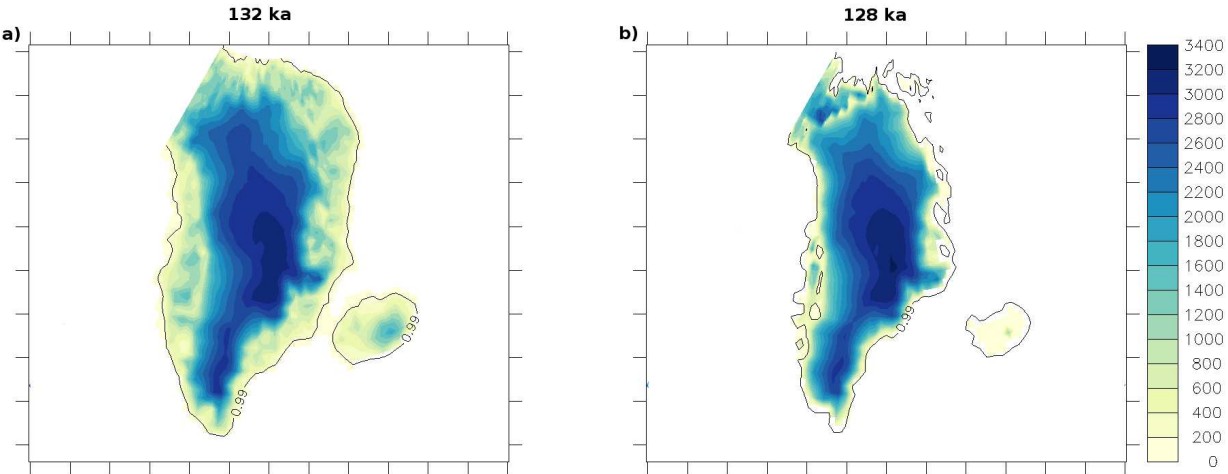

**Figure 5. Greenland ice-sheet** elevation (m) at **a)** 132 and **b)** 128 ka as simulated by the GSM (e.g. Tarasov et al., 2012). Grounding lines are in black.





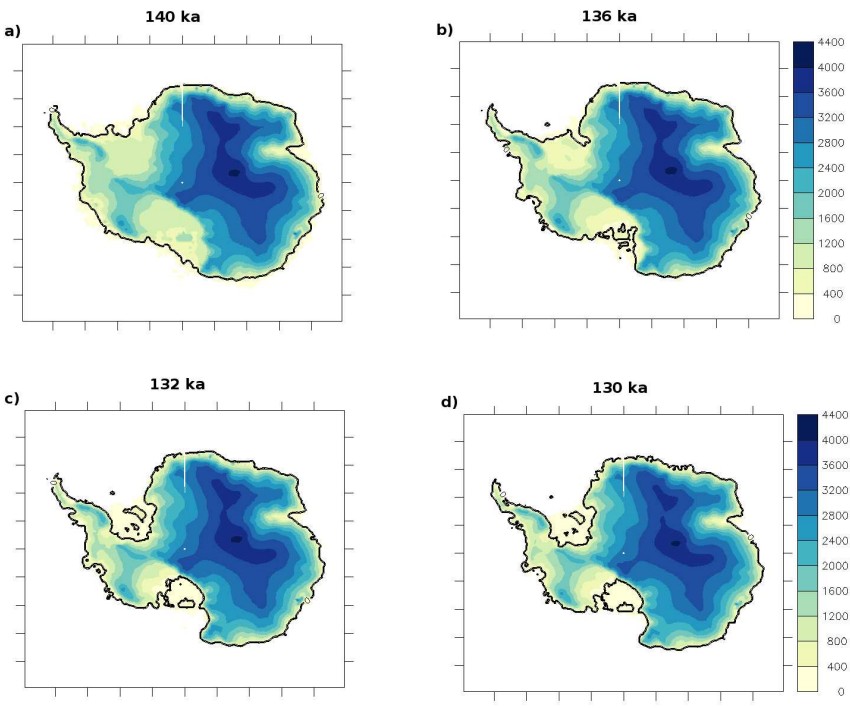

**Figure 6. Antarctic ice-sheet** elevation at **a)** 140, **b)** 136, **c)** 132 and **d)** 130 ka as simulated by the GSM (e.g. Briggs et al., 2014). Grounding lines are in black.



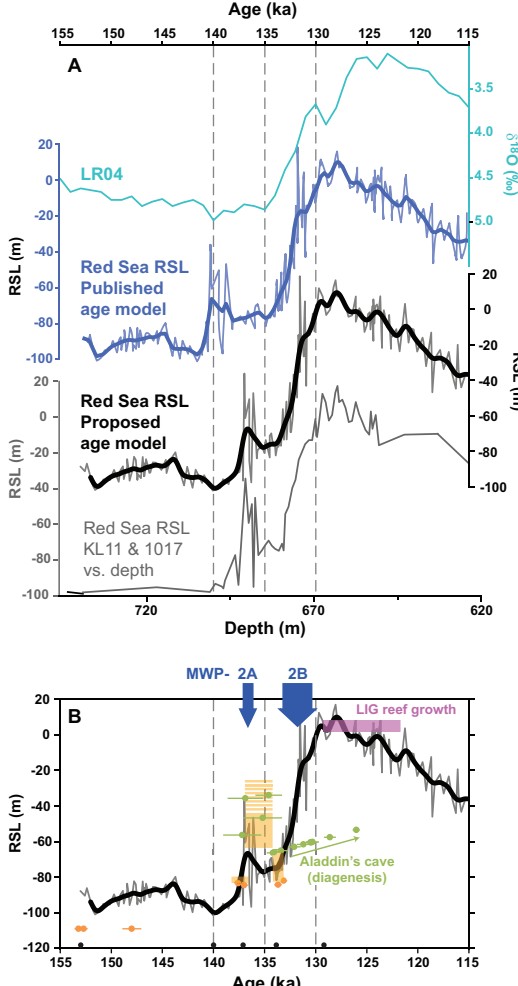

**Figure 7. Sea-level changes across TII: a)** Stacked LR04 benthic $\delta^{18}$O curve (turquoise) (Lisiecki and Raymo, 2005), Rea Sea Relative sea level (RSL) data (light blue; dark blue line, 1 kyr moving Gaussian filter) on the age model from Grant et al. (2012) and on the new age model (this study, data in grey, 1 kyr moving Gaussian filter in black); RSL curve inferred from Red sea Geo-KL11 and MD92-1017 cores on a depth scale (dark grey) (Grant et al., 2012). **b)** Tahiti (orange dots) (Thomas et al., 2009) and Huon Peninsula (green dots) (Esat et al., 1999) corals superimposed onto the Red Sea RSL curve displayed on the new age model (black). Orange bars indicate the range of the paleowater depth estimates from Tahiti corals. Note that coral data have been updated to new decay constants (Cheng et al., 2013) (Tables S3, S4). Black dots indicate the tie points defined so that the timing of MWP-2A in the Red Sea RSL is consistent with the absolutely-dated Tahiti and Huon Peninsula coral data (Table S5). The pink box represents the timing and range of RSL during the main phase of LIG reef growth (129-122 ka) (Dutton and Lambeck, 2012). Vertical dashed grey lines indicated the 140, 135 and 130 ka time intervals on both panels.



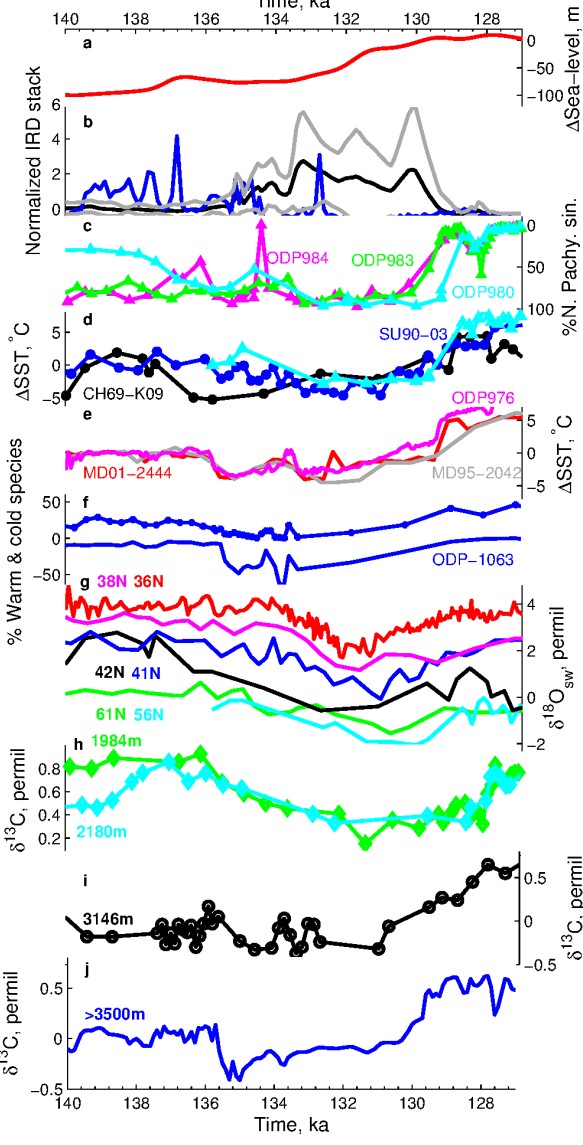

**Figure 8. Selection of RSL and North Atlantic marine sediment records: a)** Red Sea RSL record (Grant et al., 2012), with revised age-scale as described in Section 5. **b)** Normalized IRD stack (black) of cores CH69-K09 (Labeyrie et al., 1999), ODP980 (Oppo et al., 2006), ODP983 (Barker et al., 2015), ODP984 (Mokeddem et al., 2014), ODP1063 (Deaney et al., 2017) and SU90-03 (Chapman and Shackleton, 1998) with $\pm 1\sigma$ (grey envelope). Normalized IRD record of MD95-2010 (blue) (Risebrobakken et al., 2006). **c)** *N. Pachyderma sin.* (%) in cores ODP980 (cyan) (Oppo et al., 2006), ODP983 (green) (Barker et al., 2015) and ODP984 (magenta) (Mokeddem and McManus, 2016); **d)** Estimated summer SST anomalies (°C) in cores ODP980 (cyan) , CH69-K09 (black) (Labeyrie et al., 1999) and SU90-03 (blue) (Chapman and Shackleton, 1998); **e)** Uk'37 SST anomalies (°C) in cores MD01-2444 (red) (Martrat et al., 2007), ODP976 (magenta) and MD95-2042 (grey) (Martrat et al., 2014); **f)** % of warm (circles) and cold (axis reversed) species in core ODP1063 (Deaney et al., 2017); **g)** Estimated $\delta^{18}O_{sw}$ (‰) in cores ODP976 (red), MD95-2042 (magenta) (Martrat et al., 2014), SU90-03 (blue) (Chapman and Shackleton, 1998), CH69-K09 (black) (Labeyrie et al., 1999), ODP984 (green) (Mokeddem and McManus, 2016), ODP980 (cyan) (Oppo et al., 2006); Benthic $\delta^{13}C$ (‰) in cores **h)** ODP983 (green) (Lisiecki and Raymo, 2005) and ODP980 (cyan) (Oppo et al., 2006), **i)** MD95-2042 (black) (Martrat et al., 2014), **j)** a $\delta^{13}C$ stack of cores U1308, CH69-K09 and ODP1063 (Labeyrie et al., 1999; Hodell et al., 2008; Deaney et al., 2017). All the age models have been revised as described in the methods and in Govin et al. (2015).



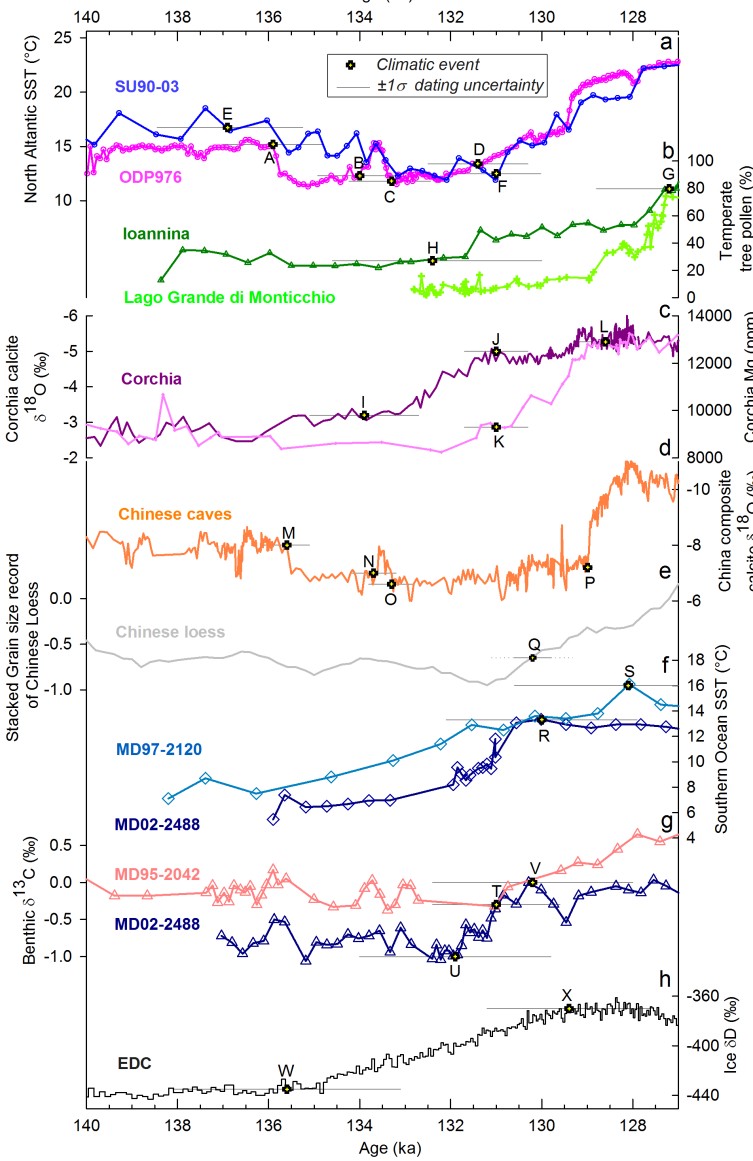

**Figure 9. Selection of paleorecords illustrating climatic and environmental changes between 140 and 127 ka. a)** SST records from North Atlantic cores ODP-976 (pink) (Martrat et al., 2014) and SU90-03 (blue) (Chapman et al., 2000). **b)** Percentage of temperature tree pollen from the Greek Ioannina sequence (dark green) (Tzedakis et al., 2003) and from the Italian Lago Grande di Monticchio (LGdM) sequence (light green) (Brauer et al., 2007). **c)** Composite speleothem calcite $\delta^{18}$O record from the Italian Corchia Cave speleothem (purple) and Corchia CD Mg concentration (Drysdale et al., 2007, 2018) **d)** Composite calcite $\delta^{18}$O record from the Chinese Hulu, Sanbao and Dongge Caves speleothems (orange) (Cheng et al., 2016). **e)** Staked grain size record of Chinese Loess (light grey) (Yang and Ding, 2014). **f)** SST records from Southern Ocean cores MD97-2120 (light blue) (Pahnke et al., 2003) and MD02-2488 (dark blue) (Govin et al., 2012). **g)** $\delta^{13}$C record from North Atlantic core MD95-2042 (light pink) (Shackleton et al., 2003) and Southern Ocean core MD02-2488 (dark blue) (Govin et al., 2012). **h)** Antarctic ice $\delta^{18}$O record from EPICA Dome C (Jouzel et al., 2007) on AICC2012 chronology (black) (Bazin et al., 2013; Veres et al., 2013). Chronologies and associated references are detailed in Table 4. Climatic events (yellow crosses) associated with $\pm 1\sigma$ dating uncertainty (horizontal grey bars) are indicated (Table 4). No details on the dating errors attached to the Chinese Loess stacked grain size record are provided in (Yang and Ding, 2014), the 1$\sigma$ error should thus be treated as an underestimated dating error.