# Peer review of "The penultimate deglaciation: protocol for PMIP4 transient numerical simulations between 140 and 127 ka"

_Climate of the Past, 2018_

## Referee Comment (RC1) · Anonymous Referee #1 · 22 Oct 2018

Menviel and colleagues describe the experimental protocol for numerical climate simulations of the penultimate deglaciation (140-127 ka) in the framework of the Paleoclimate Modelling Intercomparison Project (PMIP). The manuscript gives a comprehensive overview of the planned simulations, the required input and forcing data and paleo information that will become useful for data-model comparison later in the project.

The manuscript represents an important contribution for the modelling community to describe, organise and guide climate model experiments for the penultimate deglaciation. Next to the more technical description of the experiments, the work also gives a brief but well-informed overview of the current knowledge and scientific questions for

the climate evolution of this period. The manuscript is well written and is an interesting read for the readers of CP, even beyond the group of people directly connected to PMIP. I recommend publication of the article with minor revisions detailed below.

P1, Abstract Here you define 'thousand years before present' as 'ka', but in the rest of the text this is not used consistently. E.g. P2 L18-19 you use kyr. Consider revising for consistency.

P2 L3 Depending on the complexity of the Earth system model, some of the elements listed here as boundary conditions could be interactive components (carbon cycle, ice sheets). I suggest to make it clear that the target model configuration for these experiments are standard AOGCMs.

P2 L11 Add 'last deglaciation experiments of the' before 'PMIP4 effort'.

P2 L13 Again, clarify that this is typically O-A coupled and not including e.g. the ice sheets.

P2 L24 I can see that the earlier definition of the term 'termination' may not be very useful in the context of this effort. Nevertheless, it is not clear what the new definition really means. Maybe it is not necessary to have a precise definition, but could you try to capture the common understanding in your group. Is this e.g. from peak glacial to peak interglacial in temperature or similar?

P2 L32 While I agree that GHG and ice sheets are probably the largest factors, I am not sure other mechanisms can be excluded, like sea-ice and vegetation changes. This could be included in the sentence.

P3 L1 You could mention here how many glacial terminations have occurred in the past 450 ka.

P4 L1 Could add 'global' before CO2 to avoid confusion.

P5 Reformulate to avoid confusion: Sea-level rise is not the potential cause of HS11.

P5 L9 Specify over how many ka, or from when to when CO2 increase was sustained.

P6 L14 Is Berger (1978) the quasi-standard used by everybody in the community? Would another solution (e.g. Lasker) be accepted? Clarify.

P6 L31 Is it desired that the model configuration is the same for the proposed experiments as for the last deglaciation? Maybe this could be included as a (soft) constraint.

P7 L19 I was a bit surprised by some of the wording in the experiment description. I would e.g. interpret 'recommended' as 'other options are also OK'. I don't know if that is intended, but would suggest to carefully reconsider these formulations. I understand that on one hand the MIP cannot be too prescriptive, to not exclude specific groups or modelling approaches. On the other hand, consistent analysis across ensemble members gets very difficult when the ensemble is too diverse.

P7 L29 Will all forcing records be provided by PMIP4? I see that you have a general statement in the "Data availability" section. Maybe it could be mentioned already at an earlier state and in the main text, e.g. around here, where different records are discussed.

P8 L13 What is meant by 'Glacial geological data'? Could you give some examples?

P8 L20 Maybe 'the different' instead of 'all different'. There are not that many.

P8 L21 Also, could you be more specific on how the NHIS and GrIS are merged? How much of a difference does it make on one degree resolution to use GSM GrIS compared to what is simulated for GrIS by IcIES?

P8 L23 Be consistent with symbol GLAC1-D vs GLAC-1D.

P8 L19-29 Could you explain this better? Maybe you could start by explaining that you need TII data, but that the SL solver needs a full evolution until present day.

P8 L29 What is 'surface drainage pointer evolution'?, Explain.

P9 L3 Is it true that the ice sheet model is run on a lat-lon grid? That was surprising to me.

P9 L7 Add 'other' before 'numerical simulations'.

P9 L10-17 This paragraph could be part of an introduction in the main section 4. Consider revising.

P10 L1-10 The level of detail in this description seems a bit unbalanced compared to the limited information given for IcIES. Maybe the IcIES description could be matched to give some similar information on ice flow, parameterisations, ...

P10 L5 Add 'and bedrock geometry' after ice if that is the case.

P10 L6 What does '*partly* glacial index based' mean?

P10 L14 Could move sentence 'The last 20 kyr ...' before sentence starting 'The simulation presented ...'.

P10 L18 What is 'glacial non-floating ice volume'. Is that ice grounded out on the continental shelf?

P10 L30 Not clear what is meant here. Maybe 'The maximum sea level contribution from Greenland at 123-121 ka ...'.

P11 section 5 At first view it seems a bit out of balance to have three pages of text to describe TII sea-level evolution. Is the sea-level evolution used for anything else then the land-sea mask evolution in these simulations?

P14 L20 What is meant by 'restrictions', clarify?

P14 L30 Specify (if) what is included in the simulations. A isotope tracer?

P15 L19 It is difficult for me to imagine how land-sea masks are linearly interpolated. Please clarify.

P15 L27 How and when will these files be provided? Specify.

P16 L17 Please check and specify the units for this equation. P-E would probably be in mm/yr water equivalent, while dH/dt would probably be in ice equivalent if taken from elevation changes of the ice sheet model. What ice density is assumed in the ice sheet models? Is it consistent between IcIES and GSM?

P16 L11 There are two Goelzer et al. papers from 2016. The one that would be more appropriate to cite in this context is the one on ice sheet freshwater forcing (doi:10.5194/cp-12-1721-2016)

P17 L4 Such massive input of freshwater as given in some of the scenarios may be problematic for some of the models, especially when distributed over small areas, like river mouths. You may want to consider a plan B for such cases.

P17 L20 It is not further described what should be done in cases of negative fwf. It seems important to note that the salt flux anomaly should not be applied at the same routing locations as the positive fwf would be. Probably it should be added as a global flux if conservation is required.

P18 L27 'could be assessed', P19 L2 ' ... should be studied in detail through sensitivity experiments', P19 L5' ... sensitivity simulations forced with different dust-flux scenarios are encouraged' Is there further guidance from PMIP on these experiments, or is it up to the participants to decide these extensions? Maybe some more details could be given that support the participants in their choices?

P24 L10 Is sea-level not needed to determine changes of the land-sea mask? If not, why is there so much emphasis on this new chronology?

P24 L24 Are climate models with interactive ice sheet components not used by any of the potential participants? If they are, how would the protocol have to be adjusted to accommodate the additional capability in the best possible way?

Table 4. The caption is very difficult to read at this font size.

Figure 2 The grey EDML line is very difficult to see.

Figure 3 The caption suggests that the ice mask should be visible, but that is not the case. I see lowest bin colour everywhere.

Figure 5 Remove the 0.99 label from the grounding line.

The choice to show only 132 and 128 seems surprising. Why not show more snapshots including the beginning (140 ka). (the grid could be almost half the size!) .

Figure 6 Why not show also 134, 138 and 128 ka?

It seems that the x and y axis have different scale. Could this be improved (e.g. matlab axis equal)

Figure 7 Replace 'Rea' by 'Red' in first line.

Figure 9 Line e for Chinese loess is not well visible.

---

## Short Comment (SC1) · 22 Oct 2018

In section 3 it is described which greenhouse gas records should be used as forcing for these transient simulations across Termination II. I suggest to revise this section by refering to a recent paper I published together with colleagues, in which the greenhouse gas (GHG) records of the last 156 kyr have been compiled (Köhler et al., 2017). Interestingly, this paper is already cited in the discussion paper.

The intention of this data compilation publihsed in Köhler et al. (2017) was exactly to give transient simulation studies the best choice of GHG as forcing at hand. I recommend to use the calculated splines which are given with temporal spacing of 1 yr. For

those interested in the effect of uncertainties in the GHG there is also an uncertainty coming together with the spline of each GHG (CO2, CH4, N2O). These splines also take care of the problem that one data set might include different local maximum values than others (e.g.  $CO_2$  during the last Interglacial in Lourantou et al., 2010 versus Schneider et al., 2013).

For the problem of missing N2O data before 134 kyr BP Köhler et al. (2017) gives no alternative solution than presented here (using the LGM N2O value of 201 ppb for the times before 134 kyr). However, to avoid any artificial rapid jumps I suggest to linearly increase N2O from 201 ppb at 140 kyr to 218.74 ppb at 134.519 kyr BP, which is the oldest value of the N2O spline.

Both  $CO_2$  and  $CH_4$  undergo some abrupt changes around and before 140 kyr BP, which is in the paper suggested as the time at which the transient simulation start and for which conditions the models should spin up. However, these changes in the GHGs make it difficult to pick values for the spin up. To avoid rapid change in the GHGs I nevertheless suggest to take the values which are given by the splines at 140 kyr BP, which are 192 ppm (CO2) and 425 ppb (CH4), which differ from the values suggested in the paper (Table 1). Alternatively, one might think about taken a different time for which all boundary conditions (GHGs, orbital parameters, ice sheets) have to be taken for the spin ups (e.g. 141 kr BP during which  $CH_4$  (341 ppb) and  $CO_2$  (186 ppm) are in local minima). A second alternative would be to average both GHGs over a longer time span, e.g. across 2 kyr (from 139-141 kyr BP leading to 191 ppm (CO2) and 385 ppb ( $CH_4$ ), and linearly interpolate from those average values used during spin up and at 140 kyr BP until the onset of the pure use of the transient GHGs at 139 kyr BP. Otherwise the abrupt jumps (of the example above of +6 ppm (CO2) and +3 ppb (CH4) within 1 year) might lead to artificial responses in the simulated climate system. One might also think about averaging the GHG for some periods before 140 kyr (e.g. 140-142 kyr BP), but nevertheless one might need a linear interpolation from the average values used during spin up to first meaningful values of the transient runs to avoid
artifical jumps in the GHG radiative forcing.

Data connected with Köhler et al (2017), including raw data and final splines, are available at https://doi.org/10.1594/PANGAEA.871273.

**References**

- Köhler, P., Nehrbass-Ahles, C., Schmitt, J., Stocker, T. F., and Fischer, H.: A 156 kyr smoothed history of the atmospheric greenhouse gases CO2, CH4, and N2O and their radiative forcing, Earth Syst. Sci. Data, 9, 363-387, https://doi.org/10.5194/essd-9-363-2017, 2017.
- Lourantou, A., Lavric, J., Köhler, P., Barnola, J.-M., Paillard, D., Michel, E., Raynaud, D., and Chappellaz, J.: Constraint of the CO2 rise by new atmospheric carbon isotopic measurements during the last deglaciation, Global Biogeochemical Cycles, 24, https://doi.org/10.1029/2009GB003545, 2010.
- Schneider, R., Schmitt, J., Köhler, P., Joos, F., and Fischer, H.: A reconstruction of atmospheric carbon dioxide and its stable carbon isotopic composition from the penultimate glacial maximum to the last glacial inception, Climate of the Past, 9, https://doi.org/10.5194/cp-9-2507-2013, 2013.

---

## Short Comment (SC2) · 26 Oct 2018

I have read the manuscript and find this an effort of interest. However, I find the link to the fourth phase of the PMIP project (PMIP4) confusing at the very least.

The manuscript presented is the result of a work from a PAGES working group but is not endorsed by the PMIP4 effort nor by the deglaciation working group of PMIP4 to my knowledge. Nonetheless, the authors are presenting it as a "protocol for PMIP4 transient simulations" (title) which is incorrect. Furthermore, it has requested and apparently been granted the access to the PMIP4 Special Issue (interjournal CP/GMD)

but did not follow the guidelines of the Special Issue that requested (to my knowledge again) the inclusion of the protocol manuscripts in the GMD part of the SI and not in CP. To further complicate the understanding, if the abstract correctly states the relationships between the different projects, the body of the text is less clear.

**Abstract:**
*Here, as part of the PAGES-PMIP working group on Quaternary Interglacials, we propose a protocol to perform transient simulations of the penultimate deglaciation to complement the PMIP4 effort.* which clearly indicates that the proposed protocol is not a PMIP4 one but a complement.

**Text body:**
*We thus propose to extend the PMIP4 working group on the last deglaciation to include the penultimate deglaciation and thus create a DeglaMIP working group.* This proposal has not been discussed in the PMIP4 deglacial WG to my knowledge. It is thus rather peculiar to see this aspect claimed here.
*To facilitate the transient simulations of TII, we are providing a combined ice-sheet forcing (available on the PMIP4 portal), in which all different ice-sheets are merged.* and *Data availability. All the forcing files as well as the paleo-data described in the manuscript will be available on the PMIP4 website [link to PMIP] upon publication.* are further confusing since forcing on the PMIP4 portal should be restricted to PMIP4 protocol data.

Overall, I find that neglecting as such to clearly state what the status of this particular protocol is with regard to the official PMIP4 protocol is confusing for most climate modeling groups. I therefore think that the current manuscript should make clear that:

1. the present protocol is **not** a PMIP4 endorsed protocol. This should be made

crystal clear in the text, the manuscript should not be part of the inter-journal SI and the title should be modified to reflect that.

2. the present protocol is to be seen as complementary to PMIP4 and indeed as stated, as a potential bridge between different PMIP4 aspects. It should not be hosted on the same PMIP4 webpages or at the very least if no alternatives can be found, the pages should very clearly explain that the files are hosted by PMIP but not endorsed by PMIP (still confusing I think).

Best wishes,
Didier (R.)

---

## Short Comment (SC3) · 8 Nov 2018

My comments concern section 5 (sea level) and Figure 7. First, an important edit would be to plot the 'maximum probability' Red Sea RSL record with its 68% +/or 95% probability intervals (all Red Sea RSL data are available online). As it is, the Red Sea RSL record is plotted in Fig.7 as raw datapoints with the authors' own smoothing. This is a far less accurate representation of the original work, and rather misleading as the datapoints are from a stack of different cores and this accounts for much of the scatter. By plotting the uncertainty intervals we would be able to see where there is larger uncertainty in the Red Sea RSL record (there is indeed a large bulge at ∼136-142 ka),

hence where there is room for movement in the chronology. Outside of this bulge, the Red Sea chronology is well constrained over TII thanks to good signal agreement between the Soreq, Mediterranean, and Red Sea records (used for tuning). The authors do not note this and instead assume that there is an issue with the chronology ("Such a mismatch is likely to be related to dating uncertainties associated with the current Red Sea RSL age scale"; pg 13 ln 5). Interestingly, if the probabilistic Red Sea RSL records were plotted, then it looks like the coral data would overlap within uncertainties, with the Tahiti data overlying the first rise in RSL rather than the fall; the start of the main TII rise would also overlap within uncertainties (see attached figure, Tahiti data in orange). In other words, the timing of MWP-2A is ambiguous - either chronology (ie, the original Red Sea or the coral-adjusted Red Sea) is justifiable. This is not acknowledged by the authors. This ambiguity then begs the question of why adjust the Red Sea chronology over the MWP-2A interval. . .? I don't yet see how it is important for the remainder of the manuscript. I fully support chronological improvements if they are substantial and/or well-justified.

—————————————————————

[Figure]

**Fig. 1.** Sea level over TII

---

## Short Comment (SC4) · 12 Nov 2018

We thank Peter Köhler for his useful comment and advices.

Accordingly, we have modifed section 2 and figure 2 in the revised manuscript. Particularly:
- we now recommend using the final spline-smoothed GHG data published in Köhler et al. (2017).
- regarding the issue related to missing $N_2O$ data before 134 ka we also follow his suggestion to linearly increase $N_2O$ from 201 ppb at 140 ka to 218.74 ppb at 134.5 ka.
- regarding the greenhouse gas concentration values to use for the 140 ka spin-up, we now recommend using the averaged $CO_2$ and $CH_4$ concentrations between 139 and 141 ka, i.e. 191 ppm and 385 ppm respectively. Consequently, the $CO_2$ and $CH_4$ values between 140 and 139 ka will be linearly interpolated between the 140 ka spin-up values and the values at 139 ka from the spline-smoothed curves to avoid artificial jumps: 196.68 ppm and 287.65 ppb respectively.

Please find attached below the revised section 2 and Figure 2.

Many thanks again.

**Reference:**
Köhler, P., Nehrbass-Ahles, C., Schmitt, J., Stocker, T. F., and Fischer, H.: A 156 kyr smoothed history of the atmospheric greenhouse gases CO2, CH4 and N2O and their radiative forcing, Earth Syst. Sci. Data, 9, 363-387, https://doi.org/10.5194/essd-9-363-2017, 2017

GHG records are available solely from Antarctic ice cores across the time interval 140-127 ka (Fig. 2). LIG GHG records from the NEEM and other Greenland ice cores are affected by stratigraphic disturbances and in-situ $CO_2$, $CH_4$ and $N_2O$ production (e.g. Tschumi and Stauffer, 2000; NEEM community members, 2013). The NGRIP ice core provides a continuous and reliable $CH_4$ record but it only extends back to $\sim$123 ka (North Greenland Ice Core Project members, 2004). After a brief description of existing atmospheric $CO_2$, $CH_4$ and $N_2O$ records (below), we recommend using recent spline-smoothed GHG curves calculated from a selection of those records (Köhler et al., 2017). They have the benefit to provide continuous GHG records, with a temporal resolution of 1 yr on the commonly-used AICC2012 gas age scale (Bazin et al., 2013; Veres et al., 2013). Note that this time scale is associated with an average $1\sigma$ absolute error of $\sim$2 kyr between 140 and 127 ka.

Atmospheric $CO_2$ concentrations have been measured on the EDC and TALDICE ice cores (Fig. 2). The EDC records from Lourantou et al. (2010) and Schneider et al. (2013) agree well overall. The Schneider et al. (2013) dataset depicts a long-term $CO_2$ increase starting at $\sim$137.8 ka and ending at $\sim$128.5 ka with a centennial-scale $CO_2$ rise above the subsequent LIG $CO_2$ values, also referred to as an "overshoot". The $CO_2$ overshoot is smaller in the Schneider et al. (2013) dataset compared to a similar feature measured in Lourantou et al. (2010): while the former displays a relatively constant $CO_2$ concentration of $\sim$275 ppm between 128 and 126 ka, the latter shows a $CO_2$ decrease from 280 to 265 ppm between 128 and 126 ka. The offsets between $CO_2$ records from the same EDC core are likely related to the different air extraction techniques used in the two studies (Schneider et al., 2013). The smoothed spline $CO_2$ curve across TII we recommend using as forcing is based on those two EDC dataset and the calculation method accounts for such potential difference in local maxima (details provided in Köhler et al. (2017)).

Atmospheric $CH_4$ concentration records from Vostok, EDML, EDC and TALDICE agree well within the gas-age uncertainties attached to each core (Fig. 2). They illustrate a slow rise from $\sim$390 to 540 ppb between $\sim$137 ka and 129 ka that is followed by an abrupt increase of $\sim$200 ppb reaching maximum LIG values at $\sim$128.5 ka. Because $CH_4$ sources are located mostly in the NH, an interpolar concentration difference (IPD) between Greenland and Antarctic $CH_4$ records exists.

For instance, an IPD of $\sim$14 ppb, $\sim$34 ppb and $\sim$43 ppb is reported during the LGM, Heinrich Stadial 1 and the Bølling warming respectively (Baumgartner et al., 2012). However, without reliable $CH_4$ records from Greenland ice cores, it remains challenging to estimate the evolution of the IPD across TII. Hence, for the atmospheric $CH_4$ forcing of future TII transient simulations, we recommend using the smoothed spline $CH_4$ curve which is solely based on the EDC $CH_4$ record (Köhler et al., 2017), recognising that the values may be 1-4% lower than the actual global average.

Both $CO_2$ and $CH_4$ concentrations undergo some rapid changes around 140 ka, which is also the time when the models should spin up. To avoid possible artificial abrupt changes in the GHG we recommend using as spin-up $CO_2$ and $CH_4$ concentrations, the average values obtained for the interval 139-141 (i.e. 191 ppm for $CO_2$ and 385 ppb for $CH_4$, Table 1). Consequently, $CO_2$ and $CH_4$ changes between 140 and 139 ka provided in the forcing scenarios are linearly interpolated between the 140 ka spin-up values and those at 139 ka of 196.68 ppm for $CO_2$ and 287.65 ppb for $CH_4$. From 139 ka, the use of the spline-smoothed curves from Köhler et al. (2017) are recommended.

Atmospheric TALDICE, EDML and EDC $N_2O$ records are available between 134.5 and 127 ka (Fig. 2) (Schilt et al., 2010; Flückiger et al., 2002). From 134.5 to 128 ka, $N_2O$ levels increase from $\sim$220 to 270 ppb. Following a short decrease until $\sim$127 ka, $N_2O$ concentrations stabilise afterwards. No reliable atmospheric $N_2O$ concentrations are available beyond 134 ka as $N_2O$ concentrations measured in the air trapped in ice from the penultimate glacial maximum are affected by in-situ production related to microbial activity (Schilt et al., 2010). During the LGM (considered here as the time interval 26-21 ka), the average $N_2O$ level was $\sim$201 ppb. Assuming the LGM is an analogue for the penultimate glacial maximum, we propose a 140 ka spin-up value and $N_2O$ transient forcing curve that starts with a 201 ppb level and then linearly increases to 218.74 ppb at 134.5 ka. From 134.5 ka, we recommend using the $N_2O$ smoothed spline curve calculated by Köhler et al. (2017) and which is based on the TALDICE and EDC discrete $N_2O$ measurements.

Note that the $CO_2$ and $N_2O$ levels from the spline curves at 127 ka (274 ppm and 257 ppb) only differ from the values chosen as boundary conditions for the PMIP4 *lig127k* equilibrium experiment by 1 ppm and 2 ppb respectively (Otto-Bliesner et al., 2017; Köhler et al., 2017). The comparison is less direct for $CH_4$. Indeed a global $CH_4$ value (685 ppm) rather than an Antarctic ice core-based $CH_4$ value (e.g. $CH_4$ level of 660 ppm at 127 ka in Köhler et al. (2017)) is proposed as forcing for the *lig127k* simulations. However, this difference in global atmospheric $CH_4$ and Antarctic ice core $CH_4$ concentration is similar to the one observed during the mid-Holocene (23 ppb) (Otto-Bliesner et al., 2017; Köhler et al., 2017).

**References**

Baumgartner, M., Schilt, A., Eicher, O., Schmitt, J., Schwander, J., Spahni, R., Fischer, H., and Stocker, T. F.: High-resolution interpolar difference of atmospheric methane around the Last Glacial Maximum, Biogeosciences, 9, 3961–3977, https://doi.org/10.5194/bg-9-3961-2012, https://www.biogeosciences.net/9/3961/2012/, 2012.

5 Bazin, L., Landais, A., Lemieux-Dudon, B., Kele, H. T. M., Veres, D., Parrenin, F., Martinerie, P., Ritz, C., Capron, E., Lipenkov, V., Loutre, M.-F., Raynaud, D., Vinther, B., Svensson, A., Rasmussen, S. O., Severi, M., Blunier, T., Leuenberger, M., Fischer, H., Masson-Delmotte, V., Chappellaz, J., and Wolff, E.: An optimized multi-proxy, multi-site Antarctic ice and gas orbital chronology (AICC2012): 120-800 ka, Climate of the Past, 9, 1715–1731, 2013.

Flückiger, J., Monnin, E., Stauffer, B., Schwander, J., Stocker, T., Chappellaz, J., Raynaud, D., and Barnola, J.-M.: High-resolution Holocene $N_2O$ ice core record and its relationship with $CH_4$ and $CO_2$, Global Biogeochemical Cycles, 16, 10–1–10–8, https://doi.org/10.1029/2001GB001417, 2002.

Köhler, P., Nehrbass-Ahles, C., Schmitt, J., Stocker, T., and Fischer, H.: A 156 kyr smoothed history of the atmospheric greenhouse gases $CO_2$, $CH_4$, and $N_2O$ and their radiative forcing, Earth System Science Data, 9, 363–387, https://doi.org/10.5194/essd-9-363-2017, 2017.

Lourantou, A., Lavric, J., Kohler, P., Barnola, J.-M., Paillard, D., Michel, E., Raynaud, D., and Chappellaz, J.: Constraint of the 15 $CO_2$ rise by new atmospheric carbon isotopic measurements during the last deglaciation, Global Biogeochemical Cycles, 24, doi:10.1029/2009GB003 545, 2010.

NEEM community members: Eemian interglacial reconstructed from a Greenland folded ice core, Nature, 493, 489–494, https://doi.org/doi:10.1038/nature11789, 2013.

North Greenland Ice Core Project members: High-resolution record of Northern Hemisphere climate extending into the last interglacial 20 period, Nature, 431, 147–151, 2004.

Otto-Bliesner, B., Braconnot, P., Harrison, S., Lunt, D., Abe-Ouchi, A., Albani, S., Bartlein, P., Capron, E., Carlson, A., Dutton, A., Fischer, H., Goelzer, H., Govin, A., Haywood, A., Joos, F., Legrande, A., Lipscomb, W., Lohmann, G., Mahowald, N., Nehrbass-Ahles, C., Peterschmitt, J.-Y., Pausata, F.-R., Peterschmitt, J.-Y., Phipps, S., Renssen, H., and Zhang, Q.: The PMIP4 contribution to CMIP6 - Part 2: Two Interglacials, scientific objectives and experimental design for Holocene and Last Interglacial simulations, Geoscientific Model 25 Development, pp. 3979 – 4003, https://doi.org/10.5194/gmd-10-3979-2017, 2017.

Schilt, A., Baumgartner, M., Blunier, T., Schwander, J., Spahni, R., Fischer, H., and Stocker, T.: Glacial-interglacial and millennial-scale variations in the atmospheric nitrous oxide concentration during the last 800,000 years, Quaternary Science Reviews, 29, 182–192, https://doi.org/10.1016/j.quascirev.2009.03.011, 2010.

Schneider, R., Schmitt, J., Köhler, P., Joos, F., and Fischer, H.: A reconstruction of atmospheric carbon dioxide and its sta-30 ble carbon isotopic composition from the penultimate glacial maximum to the last glacial inception, Climate of the Past, 9, https://doi.org/10.5194/cp-9-2507-2013, 2013.

Tschumi, J. and Stauffer, B.: Reconstructing past atmospheric $CO_2$ concentration based on ice-core analyses: open questions due to in situ production of $CO_2$ in the ice, Journal of Glaciology, 46, 45–53, 2000.

Veres, D., Bazin, L., Landais, A., Kele, H. T. M., Lemieux-Dudon, B., Parrenin, F., Martinerie, P., Blayo, E., Blunier, T., Capron, E., 35 Chappellaz, J., Rasmussen, S. O., Severi, M., Svensson, A., Vinther, B., and Wolff, E.: The Antarctic ice core chronology (AICC2012): an optimized multi-parameter and multi-site dating approach for the last 120 thousand years, Climate of the Past, 9, 1733–1748, 2013.

[Figure]

**Figure 2.** Atmospheric greenhouse gas concentrations: Atmospheric trace gases through the penultimate deglaciation from Antarctic ice cores displayed on the AICC2012 chronology (Bazin et al., 2013; Veres et al., 2013). a) Atmospheric $CO_2$ concentrations from EDC (turquoise and blue) (Lourantou et al., 2010; Schneider et al., 2013) and TALDICE (green) (Schneider et al., 2013). b) Atmospheric CH4 concentration from EDC (Loulergue et al., 2008) (blue), Vostok (Petit et al., 1999) (orange), TALDICE (Buiron et al., 2011) (green) and EDML (Capron et al., 2010) (grey). c) Atmospheric $N_2O$ concentration from EDC (Flückiger et al., 2002) (blue), EDML (Schilt et al., 2010) (grey) and TALDICE (Schilt et al., 2010) (green). Transient experiments should be forced by the smoothed splines of $CO_2$, $CH_4$ and $N_2O$ concentrations as shown in black (Köhler et al, 2017). Between 140 ka and 134.5 ka, $N_2O$ should increase linearly from 201 ppb to 218.74 ppb at (dashed black line). Red crosses indicate the 140 ka spin-up values for $CO_2$, $CH_4$ and $N_2O$ concentrations (191 ppm, 385 ppb and 201 ppb, respectively).

---

## Author Comment (AC1) · 10 Dec 2018

Didier Roche's comment highlighted a miscommunication issue with respect to the inclusion of this study into PMIP4. As detailed below, we believe this issue has been resolved and this study should be an integral part of PMIP4.

*I have read the manuscript and find this an effort of interest. However, I find the link to the fourth phase of the PMIP project (PMIP4) confusing at the very least.*
We have now clarified the link to PMIP in the revised text.

*The manuscript presented is the result of a work from a PAGES working group but is not endorsed by the PMIP4 effort nor by the deglaciation working group of PMIP4 to my knowledge. Nonetheless, the authors are presenting it as a "protocol for PMIP4 transient simulations" (title) which is incorrect. Furthermore, it has requested and apparently been granted the access to the PMIP4 Special Issue (interjournal CP/GMD).*
This arose from an earlier miscommunication, which we apologise for, and has now been cleared up. We can confirm that the T2 protocol is endorsed by PMIP4 and by the deglaciation working group and therefore that the title and use of the CP/GMD special issue is appropriate. We will be transferring the manuscript to GMD since we have been made aware that this is a more appropriate journal for the experiment protocol.

**Abstract:** *'Here, as part of the PAGES-PMIP working group on Quaternary Interglacials, we propose a protocol to perform transient simulations of the penultimate deglaciation to complement the PMIP4 effort.' which clearly indicates that the proposed protocol is not a PMIP4 one but a complement.*
For clarification this has been amended to 'Here, as part of the PAGES-PMIP working group on Quaternary Interglacials, we propose a protocol to perform transient simulations of the penultimate deglaciation under the auspices of PMIP4.'

**Text body:** *'We thus propose to extend the PMIP4 working group on the last deglaciation to include the penultimate deglaciation and thus create a DeglaMIP working group.' This proposal has not been discussed in the PMIP4 deglacial WG to my knowledge. It is thus rather peculiar to see this aspect claimed here.*
Again, we apologise for the miscommunication. The tentative proposal was discussed and agreed at the PMIP4 Stockholm meeting (September 2017) and has since been circulated and agreed within the whole deglaciation working group.

*'To facilitate the transient simulations of TII, we are providing a combined ice-sheet forcing (available on the PMIP4 portal), in which all different ice-sheets are merged.' and 'Data availability. All the forcing files as well as the paleo-data described in the manuscript will be available on the PMIP4 website [link to PMIP] upon publication.' are further confusing since forcing on the PMIP4 portal should be restricted to PMIP4 protocol data.*

The above responses address this confusion and we reiterate here that the experiment is PMIP4 endorsed and therefore it is appropriate to use the PMIP4 portal.

*Overall, I find that neglecting as such to clearly state what the status of this particular protocol is with regard to the official PMIP4 protocol is confusing for most climate modeling groups. I therefore think that the current manuscript should make clear that:*

*1. the present protocol is **not** a PMIP4 endorsed protocol. This should be made crystal clear in the text, the manuscript should not be part of the inter-journal SI and the title should be modified to reflect that.*

*2. the present protocol is to be seen as complementary to PMIP4 and indeed as stated, as a potential bridge between different PMIP4 aspects. It should not be hosted on the same PMIP4 webpages or at the very least if no alternatives can be found, the pages should very clearly explain that the files are hosted by PMIP but not endorsed by PMIP (still confusing I think).*

We have clarified that the protocol is a PMIP4 protocol and changed the confusing wording (i.e. removing reference to the experiment complementing PMIP4 experiments and instead stating that it is a PMIP4 experiment); see responses above. As such the article will be part of the inter-journal SI, the current title remains appropriate (although we have added a version number), and the protocol details and boundary condition data will be hosted on the PMIP4 website. We hope that this response clears up any confusion on these issues.

---

## Referee Comment (RC2) · Anonymous Referee #2 · 14 Dec 2018

**Significance and relevance**

This manuscript refers to the penultimate deglaciation* (time interval of 13000 years; from 140 to 127 thousands of years before present, 140k-127k). The authors provide a thorough compilation –although they designate their effort as a 'non-exhaustive selection'– of their own previously published work, with records already worked and published by others. Their work has key contributions in North Atlantic meltwater flux possible scenarios and discussion on sea level and proxy record chronologies, which appear to be published here for the first time (documented in supplementary material).

While this may not appear to be very exciting to an informed reader, in the case of the penultimate deglaciation, an effort such as this is not only laudable but very necessary. Reading the manuscript, the amount of information available for the period in question may seem accessible. However, until very recently, the penultimate deglaciation was a kind of *messy* interval, due to chronological issues and the lack of resolution in proxy records. When working at decadal-to-centennial time scales, these circumstances were preventing any precise characterisation of triggers, including modulation of a sustained deglaciation a signal, amplification –e.g. through ocean, sea/land ice, vegetation feedbacks– and globalisation –e.g. through sea level rise.

* I'd suggest keeping the designation as "penultimate deglaciation" as far as possible, rather than "termination II" or "TII" or "T2" [as mentioned in the Introduction]; the latter seems restricted to a very technical, specific aspect of the wider climate it is intended to characterise.

**Categorizing the paper**

In this study, experiment protocols for transient simulations of the penultimate deglaciation [orbital, Berger1978, greenhouse gases, from Petit1999 to Köhler2017, ice-sheets, from Tarasov2012 to Briggs2014; sea level, from Stein1993 to Grant2014, bathymetry, orography and fresh-water] together with comprehensive characterisation of paleo-data for continental climates [Antarctica and Greenland ice cores, Corchia cave, Chinese caves and loess, Monticchio and Ioannina lakes] and oceanic environments [Red sea sites, ODP984, ODP983, ODP980, MD95-2010, MD95-2042, MD01-2444, ODP976, SU90-03, U1308, CH69-K09, ODP1063, MD02-2488] do conform a complete manuscript that should certainly be of interest to the reader. Both science and presentation (TEXT, 9 FIGURES and 4 TABLES; 5 SUPPLEMENTARY TABLES) are sound.

**Organisation and length of the manuscript are satisfactory**

The protocols contain sufficient detail. The main technical notes have already been addressed by REV1. Additional comments improve important aspects (greenhouse gases by P. Köhler and sea level by K. M. Grant). Authors have prepared the profiles on the most updated compatible time scales available (WD2014, AICC2012, Corchia2018, etc [see below]). Interpretations and conclusions are justified by data. The manuscript requires hardly any improvement and my overall recommendation is that the subject merits publication with only minor revision. Below are a few suggestions, grouped in four aspects:

**1. Appropriateness. Is the subject suitable for publication in CP?, or GMD?**

Authors have clear objectives when working on the subject of the penultimate deglaciation: as part of the Past Global Changes (PAGES)- Paleoclimate Modelling Intercomparison Project (PMIP) working group on Quaternary Interglacials, (i) they are setting up a protocol to perform transient simulations of the penultimate deglaciation under the auspices of PMIP, phase 4; (ii) their efforts are unbeatably connected with the "PMIP4 working group on the last deglaciation" in the short term and (iii) they are ultimately promoting a "DeglaMIP working group" in the long term (for the next phase of the Coupled Model Intercomparison Project (CMIP)? Following the publication of Ivanovic2016, at least four co-authors (R.F. Ivanovic, L.J. Gregoire, M. Kageyama, L. Tarasov) form part of the deglaciation PMIP working group to coordinate efforts to run transient simulations of the last deglaciation. Additionally, one of executive editors of the GMD journal, D. M. Roche, would appear to be interested in ensuring that this "penultimate deglaciation" initiative is well channelized through the PMIP, and maybe CMIP?

One of the co-authors of the current study (M. Kageyama) is coordinating the special issue 'PMIP4 [Climate of the Past (CP)/ Geoscientific Model Development (GMD) inter-journal SI]' for which the manuscript is intended. In my view, this special issue

is an ideal framework for the study. In this regard, the subject would be suitable for CP (for their compilation of variability in geological archives to describe the past time interval in question, from 140k to 127k). However, my advice would be, in the benefit of both authors and readers, that the manuscript be definitely transferred to the GMD journal format (designed for public discussion of description, development, and evaluation of numerical models of the Earth's system and its components, including project protocols). As a separate, technical note:

In the Acknowledgment section, the acronyms for authors PCT and RFI do not fit exactly with any author, at least as referred in the author list as it stands now under the title.

**2. Cooperation between modellers and data producers/curators, for chronologies and details in particular**

There have been previous efforts to simulate the complete interglacial sequence or part of it. The authors correctly identify them [e.g. Bakker2013, Loutre2014, Goelzer2016, Otto-Bliesner2017, Stone2016] because they have undeniable knowledge on the subject. It is not the first time they have worked on deglaciations [e.g. Menviel2011, Menviel2018, Capron2014, Capron2017; Govin2015] and about fifty percent of the authors have direct experience with PMIP4-CMIP6 protocol papers featured in the above special issue [e.g. for the past1000 [Jungclaus2017], midHolocene and last interglacial lig127k [Otto-Bliesner2017], lgm [Kageyama2017], and midPliocene-eoi400 [Haywood2016]. The novel aspect of their current efforts is to merge this previous knowledge in order to have an accurate evaluation of uncertainties and limitations when describing environmental changes of the penultimate deglaciation. Thus, this manuscript is an example of PMIP as a forum for discussion of experimental design and appropriate techniques for comparing model results with paleoclimate reconstructions. Some brief notes:

Tables S1 and S2 A manuscript in prep. [Drysdale, R., et al., Phasing between North Atlantic sea-surface temperatures and the intensification of the East Asian monsoon across Termination II, In Prep, 2018] for the Corchia speleothem in the Mediterranean is referred to for the time-scale of sites in the North Atlantic and Western Mediterranean. If the paper in not publically available at the time the present manuscript is published, I'd suggest that authors remove the reference in preparation and point to a different reference already peer-reviewed. For instance, are the age models going to change significantly from the chronology available in the recent publication Tzedakis2018, NATURE COMM, 9(1): 4235? In any case, please update references and clarify.

Figures 8 and 9 and Tables 3 and 4 The discussion about phases and chronological uncertainties is very interesting [i.e. values given are Corchia Cave records, 700 years ($2\sigma$); ODP976 up to 1600 years on average (Table S2), AICC2012 chronology, 4000 years ($2\sigma$), etc]. It is clear that the synchronisation and alignment efforts made to keep consistency will minimise these uncertainties and provide credibility to the reasoning behind Figures 8 and 9. For these figures, could the phases defined in Tables 3 and 4 be shown here somehow? Also I do not follow the meaning/significance of the letters above each record (from A to X) in Figure 9. Please clarify with appropriate explanations whether in the figure caption (perhaps not recommendable because they are complex enough) in existing Tables 3 and 4 or in an additional new table. This study can help to dig into the details of the sequence of events under discussion, not only defining two periods, one of slow deglaciation (140k, 136k and 134k) and one of rapid deglaciation (132k, 130k and 128k), but the succession of the detailed environmental variables involved. Please make the connection of these environmental changes (designed with letters?) in paleo-reconstructions with the possible scenarios of the transient simulations/sensitivity experiments considered in the study in a way understandable to a wide audience.

**3. Long term perspectives: relevance for a wider audience**

The penultimate deglaciation is a particularly interesting period to understand in view of projected climates for the current century. Given that the slow orbital parameter movements are not going to change significantly, out-scaled variables such as meltwater flux at high latitudes and greenhouse gas emissions are supposed to gain relevance. If a DeglaMIP working group is intended for the next phase of the CMIP, please give the reasons why this could be of relevance to a wider audience, including for example, wider considerations on the carbon cycle and dynamic vegetation (in particular connected with precipitation patterns), and comments on why this deglaciation led to a period warmer than the present one, with sea level likewise well above the present one.

**4. Meet Data-Stewardship standards**

This manuscript is part of a PAGES endorsed working group (QUIGS). Data stewardship is a central objective of PAGES, as part of the entire lifecycle of research, from production to archiving of data. This is also one of the reasons why PAGES has a Data Stewardship activity. Standards for the availability of modelling codes are less developed than proxy-paleo-data standards. However, they are currently being efficiently addressed at GMD (see their code and data policy). Authors would have to find an appropriate data repository (re3data.org; ZENODO? other?) and obtain a 'data citation', which is important both for scientists and funding organisations and is different from their bibliographic citation. In addition to a standard literature citation, authors need a stand-alone data citation that has to be included in the publication's reference lists. Each curated dataset used in the manuscript is required to have a unique, persistent identifier, cited to credit original data generators. For this manuscript, I'd strongly recommend not using a landing page which can easily become obsolete in the near future (with PMIP5 and the beginning of CMIP7 in 2020, etc.), but creating instead a permanent site and pointing to it in the Data Availability section.

---

## Editor Comment (EC1) · Jungclaus (Editor) · 20 Dec 2018

Having two positive review comments at hand the manuscript by Menviel et al. appears to be suitable in general for publication. However, during the discussion phase, the issue was raised that the present manuscript, as a paper describing an experimental protocol, would fit much better in the GMD section of the inter-journal SI. After some discussion with other editors and co-authors I therefore recommend that the authors seek a transfer from CP to GMD. As a protocol for a PMIP4-endorsed experiment (see authors' comments) the paper is certainly well suited for the SI. The manuscript may need some modifications to meet GMD requirements, but a new review round could be

faster if the referees serving for the CP submission would be willing to re-assess the paper in the GMD process.

---

## Author Comment (AC2) · 8 Feb 2019

Menviel and colleagues describe the experimental protocol for numerical climate simulations of the penultimate deglaciation (140-127 ka) in the framework of the Paleoclimate Modelling Intercomparison Project (PMIP). The manuscript gives a comprehensive overview of the planned simulations, the required input and forcing data and paleo information that will become useful for data-model comparison later in the project. The manuscript represents an important contribution for the modelling community to describe, organise and guide climate model experiments for the penultimate deglaciation. Next to the more technical description of the experiments, the work also gives a brief but well-informed overview of the current knowledge and scientific questions for the climate evolution of this period. The manuscript is well written and is an interesting read for the readers of CP, even beyond the group of people directly connected to PMIP. I recommend publication of the article with minor revisions detailed below.

We thank the Reviewer for their positive comment and careful review, which helped improve the manuscript. Please find our answer to comments in blue as well as suggested text changes in green.

P1, Abstract Here you define 'thousand years before present' as 'ka', but in the rest of the text this is not used consistently. E.g. P2 L18-19 you use kyr. Consider revising for consistency.
We have amended the text and are now only using ka.

P2 L3 Depending on the complexity of the Earth system model, some of the elements listed here as boundary conditions could be interactive components (carbon cycle,ice sheets). I suggest to make it clear that the target model configuration for these experiments are standard AOGCMs.
We have modified the sentence as follows:
"It is thus important to investigate, with coupled Atmosphere-Ocean General Circulation Models (AOGCMs), the climate and environmental response…"

P2 L11 Add 'last deglaciation experiments of the' before 'PMIP4 effort'.
 We have added "last deglaciation experiments"
"to complement the last deglaciation experiments suggested as part of PMIP4."

P2 L13 Again, clarify that this is typically O-A coupled and not including e.g. the ice sheets.
The sentence has been modified as follows:
"This experiment is designed for AOGCMs to assess the coupled response of the climate system to all forcings."

P2 L24 I can see that the earlier definition of the term 'termination' may not be very useful in the context of this effort. Nevertheless, it is not clear what the new definition really means. Maybe it is not necessary to have a precise definition, but could you try to capture the common understanding in your group. Is this e.g. from peak glacial to peak interglacial in temperature or similar?

In the revised manuscript we have taken out the term "Termination II". We are now only referring to the penultimate deglaciation with the acronym "PDG".

And p4, L26:
"The penultimate deglaciation (~138-128 ka, referred here as PDG), which represents the transition between the penultimate glacial period (MIS 6, also referred to as Late Saalian, 160-140 ka) and the LIG (also referred to as MIS 5e in marine sediment cores) (Govin et al., 2015),"

P2 L32 While I agree that GHG and ice sheets are probably the largest factors, I am not sure other mechanisms can be excluded, like sea-ice and vegetation changes. This could be included in the sentence.

The sentence was modified as follows:
"These amplification mechanisms are related to the large increase in atmospheric GHG concentrations (e.g. atmospheric $CO_2$ increases by 60 to 100 ppm,  Luthi et al., (2008)) (Fig. 1), the disintegration of NH ice-sheets and their associated change in albedo (Abe-Ouchi et al., 2013), as well as changes in sea-ice and vegetation cover."

P3 L1 You could mention here how many glacial terminations have occurred in the past 450 ka.
"A pervasive characteristic of the five glacial terminations of the past 450 ka"

P4 L1 Could add 'global' before CO2 to avoid confusion.
We are now referring to:
" the global atmospheric $CO_2$ concentration"

P5 Reformulate to avoid confusion: Sea-level rise is not the potential cause of HS11.
We cut the sentence in half to avoid this link.
"This is also concomitant with Heinrich Stadial 11 (HS11)…"

P5 L9 Specify over how many ka, or from when to when CO2 increase was sustained.
We have added the time range.
"…associated with a sustained atmospheric CO2 increase of 60 ppm between ~134 and 129 ka…"

P6 L14 Is Berger (1978) the quasi-standard used by everybody in the community?
Would another solution (e.g. Lasker) be accepted? Clarify.

As the PMIP4 LGM (Kageyama et al., 2017) and last deglaciation (Ivanovic et al., 2016) experiments suggest the use of insolation forcing following Berger (1978), we also suggest the use of Berger (1978) to stay consistent with other experiments.

P6 L31 Is it desired that the model configuration is the same for the proposed experiments as for the last deglaciation? Maybe this could be included as a (soft) constraint.

Thank you, that's a good point. We have added this at the beginning of Section 6:

"To maximise the use of the transient simulations, it is suggested to perform the transient simulations of the penultimate and last deglaciations with the same version of the climate model."

P7 L19 I was a bit surprised by some of the wording in the experiment description. I would e.g. interpret 'recommended' as 'other options are also OK'. I don't know if that is intended, but would suggest to carefully reconsider these formulations. I understand that on one hand the MIP cannot be too prescriptive, to not exclude specific groups or modelling approaches. On the other hand, consistent analysis across ensemble members gets very difficult when the ensemble is too diverse.

We understand the point of the reviewer. We have adjusted our terminology so that it is clear that some forcings "should be used" (e.g. insolation, GHG, continental ice-sheets). However, for the flow of the text, we still use the term "recommend", which should be interpreted as this is what should be done.

P7 L29 Will all forcing records be provided by PMIP4? I see that you have a general statement in the "Data availability" section. Maybe it could be mentioned already at an earlier state and in the main text, e.g. around here, where different records are discussed.
All the forcing files will be available on the PMIP4 website. This is also mentioned at the beginning of section 4.

P8 L13 What is meant by 'Glacial geological data'? Could you give some examples?
We have added : "(e.g. glacial deposits, glacial striations...) "

P8 L20 Maybe 'the different' instead of 'all different'. There are not that many.
This was amended.

P8 L21 Also, could you be more specific on how the NHIS and GrIS are merged?
How much of a difference does it make on one degree resolution to use GSM GrIS compared to what is simulated for GrIS by IcIES?
IcIES was not designed to simulate the evolution of the Greenland ice-sheet. As such, experts in both GrIS and IcIES suggested it was best to use GrIS and merge it with the North American and Eurasian ice-sheets simulated by IcIES as well as with the Antarctic ice-sheet model.  The merger involves no extra smoothing (beyond that inherent in the GIA solver which involves transformation to spherical harmonics). The merger involves a simple masking operation with the mask boundary through Nares Strait, Baffin Bay, Davis Strait, and the Labrador Sea. Examination of the resultant topography shows small merger artifacts around Nares Strait ranging to a few hundred metres in elevation difference.
While the climatic impact of using the Greenland ice-sheet evolution as simulated by IcIES instead of the one simulated by the GSM GrIS should be mostly regional, we prefer to provide the best ice-sheet product we can.

P8 L23 Be consistent with symbol GLAC1-D vs GLAC-1D.
Only GLAC-1D is now used.

P8 L19-29 Could you explain this better? Maybe you could start by explaining that you need TII data, but that the SL solver needs a full evolution until present day.
We have added at the beginning of the paragraph:
"As the sea-level solver assumes an equilibrium initial condition, the simulations start at the previous interglacial. As is standard, the solver also requires present-day ice-sheet histories to bias correct against present-day observed topography. Thus, a full 240 ka ice-sheet history is required."

P8 L29 What is 'surface drainage pointer evolution'?, Explain.
This was rephrased as:
"to extract the relevant surface drainage pointer field for each time-slice. This will indicate in which ocean grid cell each terrestrial grid cell will drain into."

P9 L3 Is it true that the ice sheet model is run on a lat-lon grid? That was surprising to me.
Yes, the paleo ice-sheet model runs for North America and Eurasia used here are on lat/lon grids.

P9 L7 Add 'other' before 'numerical simulations'.
This was added.

P9 L10-17 This paragraph could be part of an introduction in the main section 4. Consider revising.
We thank the reviewer for this suggestion, which we have carefully considered. However, we think that the text of the ice-sheet section is more succinct with a direct comparison between simulated and reconstructed ice-sheet changes as currently presented.

P10 L1-10 The level of detail in this description seems a bit unbalanced compared to the limited information given for IcIES. Maybe the IcIES description could be matched to give some similar information on ice flow, parameterisations, ...
We have added to the IcIES description:
"IcIES uses the shallow ice approximation and computes the evolution of grounded ice but not floating ice shelves. The sliding velocity is related to the gravitational driving stress according to Payne (1999) and basal sliding only occurs when the basal ice is at the pressure melting point."

P10 L5 Add 'and bedrock geometry' after ice if that is the case.
The sentence was changed to:
"Model runs start at 240 ka with present-day ice and bedrock geometry and with an ice and bed temperature field ..."

P10 L6 What does '*partly* glacial index based' mean?
The climate field is a composite of two glacial indexed approaches and temperature fields from the 2D energy balance climate model (which does not involve glacial indices).
The text now reads:
"The model is then forced from 240 ka until 0 ka, with a climate forcing that is partly glacial index based, using a composite of a glaciological inversion of the GISP II regional

temperature change (for the last 40 ka) and the synthetic Greenland $\delta^{18}O$ curve that was deduced from the Antarctic EDC isotopic record assuming a thermal bipolar seesaw pattern (Barker et al., 2011). The climate forcing also includes 2-way coupled 2D energy balance climate model (Tarasov et al., 1997) to capture radiative changes."

P10 L14 Could move sentence 'The last 20 kyr ...' before sentence starting 'The simulation presented ...'.
This was changed.

P10 L18 What is 'glacial non-floating ice volume'. Is that ice grounded out on the continental shelf?
Yes, this was changed to:
"glacial grounded ice volume"

P10 L30 Not clear what is meant here. Maybe 'The maximum sea level contribution from Greenland at 123-121 ka ...'.
We have deleted this sentence.

P11 section 5 At first view it seems a bit out of balance to have three pages of text to describe TII sea-level evolution. Is the sea-level evolution used for anything else then the land-sea mask evolution in these simulations?

We are aware that this section is quite long compared to others. However, this is justified by the fact that we are presenting a revised age model for the Red Sea record, rather than just using what was published. Details are necessary to guide the reader, as highlighted by Dr. K. Grant's comment, which asked for additional justifications on the chronology.
The sea level constraints described in that section are amongst the only (indirect) observational constraints that we have regarding ice sheet melting across TII. The sea-level evolution is used to infer changes in the meltwater forcing. This is a crucial parameter in these simulations as meltwater will impact the ocean circulation and therefore climate, the global carbon cycle... It is important that the meltwater scenario gives a reasonable climate evolution across the deglaciation, but also that meltwater input stays within the probabilistic sea-level evolution. Finally, we think that this sea-level section should be of interest to a broad audience.
Nevertheless, we have tried to tighten the text in this section.

P14 L20 What is meant by 'restrictions', clarify?
We deleted 'restrictions'.

P14 L30 Specify (if) what is included in the simulations. A isotope tracer?
The sentence was expanded to be clearer:
"Furthermore, if oxygen isotopes are included in the simulations, the ocean mean $\delta^{18}O$ should be initialized at 1 ‰ and if a carbon cycle model is included, the global mean alkalinity content should be increased by about 80 µmol/L. "

P15 L19 It is difficult for me to imagine how land-sea masks are linearly interpolated. Please clarify.
The sentence refers to the ice mask and topography (L.17), and we are talking about a time interpolation.

P15 L27 How and when will these files be provided? Specify.
We have added that the files will be available from the PMIP4 data repository when the manuscript will be accepted for publication.
"Topographically-self consistent drainage routing maps will be provided on the PMIP4 data repository. "

P16 L17 Please check and specify the units for this equation. P-E would probably be in mm/yr water equivalent, while dH/dt would probably be in ice equivalent if taken from elevation changes of the ice sheet model. What ice density is assumed in the ice sheet models? Is it consistent between IcIES and GSM?

The units were given and the equation and text slightly modified to be clearer:
"the provided downslope routing fields to route the water flux (fwf, cm/yr) from each grid cell:
$fwf = (P-E)_{GCM} - (dH/dt)_{ice-sheet}*0.91$, with P for precipitation (cm/yr), E for evaporation (cm/yr) and $(dH/dt)_{ice-sheet}$ (cm/yr, assuming an ice density of 0.91 g/cm$^3$) the change in ice-sheet thickness over time as described in the ice-sheet forcing files."

P16 L11 There are two Goelzer et al. papers from 2016. The one that would be more appropriate to cite in this context is the one on ice sheet freshwater forcing (doi:10.5194/cp-12-1721-2016)
The citation was amended.

P17 L4 Such massive input of freshwater as given in some of the scenarios may be problematic for some of the models, especially when distributed over small areas, like river mouths. You may want to consider a plan B for such cases.
We have amended the description of the location of the freshwater input on p16, L.14:
"As much as possible, and for all scenarios, meltwater should be added in the appropriate locations to match the evolution of the ice sheets. Freshwater can be added over an appropriate ocean area close to the disintegrating ice-sheet, or a self-consistent paleo surface drainage forcing could also be implemented."

P17 L20 It is not further described what should be done in cases of negative fwf. It seems important to note that the salt flux anomaly should not be applied at the same routing locations as the positive fwf would be. Probably it should be added as a global flux if conservation is required.
We have added p16, L. 20:
"In case of negative meltwater forcing, the artificial salt flux addition should be spread globally over the ocean."

P18 L27 'could be assessed', P19 L2 ' ... should be studied in detail through sensitivity experiments', P19 L5' ... sensitivity simulations forced with different dust-flux scenarios are

encouraged' Is there further guidance from PMIP on these experiments, or is it up to the participants to decide these extensions? Maybe some more details could be given that support the participants in their choices?

In this manuscript, we are recommending one main experiment (described in Table 1) to be performed, and which should be similar across models. Depending on their resources, participants are encouraged to run sensitivity experiments. It is however up to each group to decide which sensitivity experiment to run, depending on the scientific question they are interested in.

P24 L10 Is sea-level not needed to determine changes of the land-sea mask? If not, why is there so much emphasis on this new chronology?

The changes in sea-level across the deglaciation are used to infer changes in the freshwater forcing. Freshwater inputs will significantly affect the deglacial climate history and it is quite important to stay within the sea-level estimates.

P24 L24 Are climate models with interactive ice sheet components not used by any of the potential participants? If they are, how would the protocol have to be adjusted to accommodate the additional capability in the best possible way?

Climate models with interactive ice sheet components are great tools to study the Earth system evolution across deglaciations. Transient simulations performed with Earth system models, which include an interactive ice-sheet component should follow the suggested insolation and greenhouse gases forcings.

The last sentence of the paragraph was reformulated as follows:

"The ultimate goal would be to perform transient simulations of the penultimate deglaciation with Earth system models that include interactive ice-sheets and carbon cycle components. But models might not be quite ready for such a task yet."

Table 4. The caption is very difficult to read at this font size.
The font of the caption was increased.

Figure 2 The grey EDML line is very difficult to see.
The EDML data is now shown in pink.

Figure 3 The caption suggests that the ice mask should be visible, but that is not the case. I see lowest bin colour everywhere.
The caption simply states that the topography is shown only when the ice-mask is greater than 0.5 (i.e. non ice-sheet related topography is not shown).

Figure 5 Remove the 0.99 label from the grounding line.
The choice to show only 132 and 128 seems surprising. Why not show more snapshots including the beginning (140 ka). (the grid could be almost half the size!) .
We are now showing the Greenland ice-sheet as simulated at 140 ka, 130 ka, 129 ka and 128 ka. As can be seen in this figure as well as in the time evolution of changes in Greenland ice mass (Fig. 4c), there is very little change between 140 and 130 ka.

Figure 6 Why not show also 134, 138 and 128 ka? It seems that the x and y axis have different scale. Could this be improved (e.g. matlab axis equal)

Figure 6 now displays the Antarctic ice-sheet evolution at 140 ka, 136 ka, 134 ka, 132 ka, 130 ka and 128 ka. These timeslices are similar to the ones shown in Figure 3 for the Northern Hemispheric icesheet. In addition, the x/y ratio was amended.

Figure 7 Replace 'Rea' by 'Red' in first line.

This was amended.

Figure 9 Line e for Chinese loess is not well visible

The color of Chinese loess in Fig. 9 was changed.

---

## Author Comment (AC3) · 8 Feb 2019

Response to comment by K. Grant

We thank K. Grant for her helpful comments, which helped improve the manuscript. Please find our answer to comments in blue as well as suggested text changes in green.

*First, an important edit would be to plot the 'maximum probability' Red Sea RSL record with its 68% +/or 95% probability intervals (all Red Sea RSL data are available online). As it is, the Red Sea RSL record is plotted in Fig.7 as raw datapoints with the authors' own smoothing.*

Clarification: In Figure 7, we plotted the raw data and the smoothed curve (RSL_smooth) provided in the supplementary material to Grant et al. 2012, not our own smoothing.
We have now added the 95% probability intervals as requested.

*Interestingly, if the probabilistic Red Sea RSL records were plotted, then it looks like the coral data would overlap within uncertainties, …. Outside of this bulge, the Red Sea chronology is well constrained over TII thanks to good signal agreement between the Soreq, Mediterranean, and Red Sea records (used for tuning). The authors do not note this and instead assume that there is an issue with the chronology ("Such a mismatch is likely to be related to dating uncertainties associated with the current Red Sea RSL age scale"; pg 13 ln 5)*

Thank you for this useful comment.  Our explanation indeed could have been more detailed to explain the justification for our treatment of the data and we welcome the opportunity to elaborate on this here. In the supplementary information of Grant et al. (2012, *Nature*), the authors note that the TII transition is more ambiguous and state that the tie points between delta-18Opac from LC21 (Mediterranean core) and the RSL reconstruction from the Red Sea were chosen as follows:

"It is common practice, when graphically correlating records, to anchor them at the mid-point of corresponding transitions, rather than using peaks or troughs in the records. We follow this approach, but make one exception for the tie-point at the base of termination II (main-text Fig. 1). We chose this position (at 136 ka) because an unambiguous tie-point is lacking over the transition due to the different step-wise structures of the two records; the records are much more similar at the base of the transition which means that we can more confidently assign a tie-point here."

Hence, the authors note that the structure of this interval makes it difficult to establish a confident tie point. There are differences when comparing the details of structure in the Soreq stack -LC21 d18O ruber – LC21 d18O pac – Red Sea RSL that make this correlation challenging.  In the revision of the chronology that we propose, the alignment of the rapid TII transition is not significantly affected.  Instead the change is proposed in a part of the LC21 d18Opac record where there is not much variability occurring.

Dr. Grant points out that some of the coral age-elevation data do lie within the 95% probability intervals of the RSL chronology published by Grant et al., (2012). Beyond this observation, and

more importantly, the timing of a SL reversal (MWP-2A) that is evident from the sedimentary observations at both Huon and Tahiti (see manuscript text) does not agree with the timing of the same event in the Red Sea using the Grant et al. (2012, Nature) chronology. The reason, therefore, that the revised chronology is proposed is to provide a better agreement between the absolutely dated (U-Th) corals to the Red Sea RSL. The basis of our decision to adjust the Red Sea chronology is (1) that the two U-Th dated coral records agree on the timing of MWP-2A and also (2) given the potential ambiguities of the TII tie point that was used to ultimately transfer the Soreq stack chronology to the Red Sea RSL. Figure 7 also shows that this adjustment coincidentally causes our new, proposed Red Sea chronology to align with the raw depth data. There is no reason to believe that a linear depth-age relationship should hold here, but it is may not be coincidental that our correction (based on the coral data) restores this alignment.

Finally, we submit that the MWP-2A (or sea level reversal) is unequivocal in the sedimentary record at Huon and Tahiti and cannot just be attributed to scatter in data points. The coral terrace at Huon must have been constructed under a higher sea level and earlier time than the head corals in Aladdin's cave (that is cut into the terrace) due to basic geologic principles of superposition and cross-cutting relationships. Additionally, the lithofacies and the benthic foraminiferal assemblage in the Tahiti cores provide evidence that paleowater depth deepened during the interval that is bounded by dates on shallow-water corals on either side of the MWP-2A event in the Tahiti cores. Hence the sedimentary evidence (not age-elevation data) are the primary observations that argue for a sea level reversal during the TII transition.

The importance of this adjustment in the chronology is that it compresses the overall duration of TII in terms of sea-level rise and ice sheet decay, and helps to better constrain the relative timing and leads and lags between other components of the climate system.

In the revised version, we have now added further justifications for the revised chronology and we state that the proposed reconstruction is still subject to debate:

"When considering the 95% probabilistic intervals of the Red Sea RSL reconstruction on the chronology from Grant et al. (2012), an overlap is observed with the coral data over the MWP-2A interval, within the stated uncertainties. Still, both coral datasets suggest that MWP-2A occurs several millennia later (i.e. ~135-134 ka) than in the Red Sea RSL reconstruction. This mismatch is likely to be related to the difficulty to precisely anchor the dating of the current Red Sea RSL age scale over this interval (as also discussed in the supplementary information of Grant et al. (2012)). Hence, we propose a revised chronology for the Red Sea RSL record in order to provide a better agreement with the absolutely-dated corals. Given the potential ambiguities of the tie point defined in Grant et al. (2012) to stretch the depth scale across this interval, we find it reasonable to adjust it such that the timing of MWP-2A is more consistent with the absolute ages provided by the Tahiti and Huon Peninsula coral data. "

"This revised chronology is still attached to large uncertainties given the limits of the datasets."

---

## Author Comment (AC4) · 8 Feb 2019

We thank the Reviewer for their positive comment and careful review, which helped improve the manuscript. Please find our answer to comments in blue as well as suggested text changes in green.

Significance and relevance

This manuscript refers to the penultimate deglaciation* (time interval of 13000 years; from 140 to 127 thousands of years before present, 140k-127k). The authors provide a thorough compilation –although they designate their effort as a 'non-exhaustive selection'– of their own previously published work, with records already worked and published by others. Their work has key contributions in North Atlantic meltwater flux possible scenarios and discussion on sea level and proxy record chronologies, which appear to be published here for the first time (documented in supplementary material). While this may not appear to be very exciting to an informed reader, in the case of the penultimate deglaciation, an effort such as this is not only laudable but very necessary. Reading the manuscript, the amount of information available for the period in question may seem accessible. However, until very recently, the penultimate deglaciation was a kind of messy interval, due to chronological issues and the lack of resolution in proxy records. When working at decadal-to-centennial time scales, these circumstances were preventing any precise characterisation of triggers, including modulation of a sustained deglaciation a signal, amplification –e.g. through ocean, sea/land ice, vegetation feedbacks– and globalisation –e.g. through sea level rise.

* I'd suggest keeping the designation as "penultimate deglaciation" as far as possible, rather than "termination II" or "TII" or "T2" [as mentioned in the Introduction]; the latter seems restricted to a very technical, specific aspect of the wider climate it is intended to characterise.

We thank the Reviewer for their positive comment and for putting our work into context. In the revised manuscript we have taken out the term "Termination II". We are now only referring to the penultimate deglaciation with the acronym "PDG".

And p4, L26:
"The penultimate deglaciation (~138-128 ka, referred here as PDG), which represents the transition between the penultimate glacial period (MIS 6, also referred to as Late Saalian, 160-140 ka) and the LIG (also referred to as MIS 5e in marine sediment cores) (Govin et al., 2015),"

**Categorizing the paper**
In this study, experiment protocols for transient simulations of the penultimate deglaciation [orbital, Berger1978, greenhouse gases, from Petit1999 to Köhler2017, ice-sheets, from Tarasov2012 to Briggs2014; sea level, from Stein1993 to Grant2014, bathymetry, orography and fresh-water] together with comprehensive characterisation of paleodata for continental climates [Antarctica and Greenland ice cores, Corchia cave, Chinese caves and loess, Monticchio and Ioannina lakes] and oceanic environments [Red sea sites, ODP984, ODP983, ODP980, MD95-2010, MD95-2042, MD01-2444, ODP976,

SU90-03, U1308, CH69-K09, ODP1063, MD02-2488] do conform a complete manuscript that should certainly be of interest to the reader.

Both science and presentation (TEXT, 9 FIGURES and 4 TABLES; 5 SUPPLEMENTARY TABLES) are sound.

We thank the Reviewer for their positive comment.

**Organisation and length of the manuscript are satisfactory**
The protocols contain sufficient detail. The main technical notes have already been addressed by REV1. Additional comments improve important aspects (greenhouse gases by P. Köhler and sea level by K. M. Grant). Authors have prepared the profiles on the most updated compatible time scales available (WD2014, AICC2012, Corchia2018, etc [see below]). Interpretations and conclusions are justified by data. The manuscript requires hardly any improvement and my overall recommendation is that the subject merits publication with only minor revision. Below are a few suggestions, grouped in four aspects:

**1. Appropriateness. Is the subject suitable for publication in CP?, or GMD?**
Authors have clear objectives when working on the subject of the penultimate deglaciation: as part of the Past Global Changes (PAGES)- Paleoclimate Modelling Intercomparison Project (PMIP) working group on Quaternary Interglacials, (i) they are setting up a protocol to perform transient simulations of the penultimate deglaciation under the auspices of PMIP, phase 4; (ii) their efforts are unbeatably connected with the "PMIP4 working group on the last deglaciation" in the short term and (iii) they are ultimately promoting a "DeglaMIP working group" in the long term (for the next phase of the Coupled Model Intercomparison Project (CMIP)? Following the publication of Ivanovic2016, at least four co-authors (R.F. Ivanovic, L.J. Gregoire, M. Kageyama, L. Tarasov) form part of the deglaciation PMIP working group to coordinate efforts to run transient simulations of the last deglaciation. Additionally, one of executive editors of the GMD journal, D. M. Roche, would appear to be interested in ensuring that this "penultimate deglaciation" initiative is well channelized through the PMIP, and maybe CMIP? One of the co-authors of the current study (M. Kageyama) is coordinating the special issue 'PMIP4 [Climate of the Past (CP)/ Geoscientific Model Development (GMD) inter-journal SI]' for which the manuscript is intended. In my view, this special issue is an ideal framework for the study. In this regard, the subject would be suitable for CP (for their compilation of variability in geological archives to describe the past time interval in question, from 140k to 127k). However, my advice would be, in the benefit of both authors and readers, that the manuscript be definitely transferred to the GMD journal format (designed for public discussion of description, development, and evaluation of numerical models of the Earth's system and its components, including project protocols).

The manuscript will be withdrawn from Climate of the Past and a revised version will be submitted to Geoscientific Model Development.

As a separate, technical note: In the Acknowledgment section, the acronyms for authors PCT and RFI do not fit exactly with any author, at least as referred in the author list as it stands now under the title.

Names in the author list have been amended.

**2. Cooperation between modellers and data producers/curators, for chronologies and details in particular**

There have been previous efforts to simulate the complete interglacial sequence or part of it. The authors correctly identify them [e.g. Bakker2013, Loutre2014, Goelzer2016, Otto-Bliesner2017, Stone2016] because they have undeniable knowledge on the subject. It is not the first time they have worked on deglaciations [e.g. Menviel2011, Menviel2018, Capron2014, Capron2017; Govin2015] and about fifty percent of the authors have direct experience with PMIP4-CMIP6 protocol papers featured in the above special issue [e.g. for the past1000 [Jungclaus2017], midHolocene and last interglacial lig127k [Otto-Bliesner2017], lgm [Kageyama2017], and midPliocene-eoi400 [Haywood2016]. The novel aspect of their current efforts is to merge this previous knowledge in order to have an accurate evaluation of uncertainties and limitations when describing environmental changes of the penultimate deglaciation. Thus, this manuscript is an example of PMIP as a forum for discussion of experimental design and appropriate techniques for comparing model results with paleoclimate reconstructions.

Some brief notes:
Tables S1 and S2 A manuscript in prep. [Drysdale, R., et al., Phasing between North Atlantic sea-surface temperatures and the intensification of the East Asian monsoon across Termination II, In Prep, 2018] for the Corchia speleothem in the Mediterranean is referred to for the time-scale of sites in the North Atlantic and Western Mediterranean. If the paper in not publically available at the time the present manuscript is published, I'd suggest that authors remove the reference in preparation and point to a different reference already peer-reviewed. For instance, are the age models going to change significantly from the chronology available in the recent publication Tzedakis2018, NATURE COMM, 9(1): 4235? In any case, please update references and clarify.
We have removed the reference to the manuscript of Drysdale et al., in preparation and instead are pointing to Tzedakis et al., (2018).

"For this current study, we carried out high-resolution measurements of d18O and Mg across the TII section of CD3, and anchored the CD3 chronology to CC5 by synchronising their d18O profiles. This enabled the Mg series of CD3 to be placed on the U-Th chronology of the 2018 Corchia Cave stalagmite stack (CCSS18, Tzedakis et al., 2018). "

Figures 8 and 9 and Tables 3 and 4 The discussion about phases and chronological uncertainties is very interesting [i.e. values given are Corchia Cave records, 700 years (2σ); ODP976 up to 1600 years on average (Table S2), AICC2012 chronology, 4000 years (2σ), etc]. It is clear that the synchronisation and alignment efforts made to keep consistency will minimise these uncertainties and provide credibility to the reasoning behind Figures 8 and 9. For these figures, could the phases defined in Tables 3 and 4 be shown here somehow?
The phases will be added as bands of colour in figures 8 and 9.

Also I do not follow the meaning/significance of the letters above each record (from A to X) in Figure 9. Please clarify with appropriate explanations whether in the figure caption (perhaps not recommendable because they are complex enough) in existing Tables 3 and 4 or in an additional new table.

The letters [A] to [X] refer to the major changes identified in the paleoclimatic records obtained through RAMPFIT or BREAKFIT. This is indicated in the legend of Table 3. This sentence was also added to the legend of Figure 9 for clarity.

This study can help to dig into the details of the sequence of events under discussion, not only defining two periods, one of slow deglaciation (140k, 136k and 134k) and one of rapid deglaciation (132k, 130k and 128k), but the succession of the detailed environmental variables involved. Please make the connection of these environmental changes (designed with letters?) in paleo-reconstructions with the possible scenarios of the transient simulations/sensitivity experiments considered in the study in a way understandable to a wide audience.

In section 8.3, we have added some new paragraphs describing in more details the environmental changes occurring during the deglaciation:

"Little change occurs until the beginning of phase 1 at ~136.4 ka, after which a cooling phase is identified in a few records of the North Atlantic (Fig. 8 and Fig. 9a, major change [A]). This also corresponds to reduced monsoon activity as recorded in Chinese speleothems (Fig. 9d [M]), and the initiation of the Antarctic warming (Fig. 9h [W]).

The short-lived warming event in the North Atlantic associated with phase 2 (Fig. 9a [B]) is also identified in the Chinese speleothems as a slightly wetter interval (Fig. 9d [N]). Other environmental records might not have the necessary resolution to record this multi-centennial-scale event.

The main phase of HS11, corresponding to phase 3, is associated with meltwater input and cold conditions in the North Atlantic (Fig. 9a [C] and 9c [I]), dry conditions over Europe (Fig. 9b) and Asia (Fig. 9d, interval between [O] and [P]), and warmer conditions at high southern latitudes (Fig. 9f, h).

The end of HS11 (phase 4) associated with a pause in the meltwater input (Fig. 9c [J]) and progressively warmer conditions in the North Atlantic and southern Europe (Fig. 9a [D, F] and 9c [K]) corresponds to a strengthening of the Asian monsoon (Fig. 9d [P] and 9e [Q]), and maximum warmth at high southern latitudes (Fig. 9f [R, S] and 9h [X]).

Interglacial conditions in atmospheric $CO_2$ and $CH_4$ as well as North Atlantic temperatures and ventilation are attained at about 128.5 ka (Fig. 8), which is also associated with warm and wet conditions in southern Europe (Fig. 9b [G], and 9c [L])."

**3. Long term perspectives: relevance for a wider audience**

The penultimate deglaciation is a particularly interesting period to understand in view of projected climates for the current century. Given that the slow orbital parameter movements are not going to change significantly, out-scaled variables such as meltwater flux at high latitudes and greenhouse gas emissions are supposed to gain relevance. If a DeglaMIP working group is intended for the next phase of the CMIP, please give the reasons why this could be of relevance to a wider audience, including for example, wider considerations on the carbon cycle and dynamic vegetation (in particular connected with

precipitation patterns), and comments on why this deglaciation led to a period warmer than the present one, with sea level likewise well above the present one.

We are setting the context for a wider audience in the abstract, where we are now adding a reference to meltwater fluxes:

"Considering the transient nature of the Earth system, the LIG climate and ice-sheets evolution were certainly influenced by the changes occurring during the penultimate deglaciation. It is thus important to investigate, with coupled Atmosphere-Ocean General Circulation Models (AOGCMs), the climate and environmental response to the large changes in boundary conditions (i.e. orbital configuration, atmospheric greenhouse gas concentrations, ice sheet geometry, and associated meltwater fluxes) occurring during this time interval."

A reference to vegetation changes was added in the Introduction:
"It is also crucial to comprehend the subsequent impacts of continental ice-sheets disintegration on the oceanic circulation and thus the climate, the terrestrial vegetation and the carbon-cycle system."

We have added in the Introduction, p2, Line 24:
"These long glacial periods were followed by relatively rapid multi-millennial-scale warmings into an interglacial state. These glacial-interglacial transitions represent the largest natural global warming and large-scale climate reorganisations across the Quaternary. Hence, they provide a great opportunity to study the interaction between the different components of the Earth System and climate sensitivity to changes in radiative forcing. "

The impact of the penultimate deglaciation on the climate and sea-level of the LIG is also mentioned in the Introduction:
"Transient simulations of the penultimate deglaciation could also help to understand the climate and sea-level highstand occurring during the LIG."

In the conclusion, we have added a reference to changes in vegetation cover and marine ecosystem:
"Transient simulations performed with Earth-system models that include a dynamic vegetation and a global carbon cycle model, would be particularly useful in assessing the impact of climate change on vegetation cover and on marine ecosystems."

We have reformulated the sentence p25, line 24:
"Transient deglacial simulations and associated model/paleo-proxy comparisons provide a great opportunity to understand the drivers and processes involved in one of the largest natural global warming period of the Quaternary."

**4. Meet Data-Stewardship standards**

This manuscript is part of a PAGES endorsed working group (QUIGS). Data stewardship is a central objective of PAGES, as part of the entire lifecycle of research, from production to archiving of data. This is also one of the reasons why PAGES has a Data Stewardship activity. Standards for the availability of modelling codes are less developed than proxy-paleo-data standards. However, they are currently being efficiently addressed at GMD

(see their code and data policy). Authors would have to find an appropriate data repository (re3data.org; ZENODO? other?) and obtain a 'data citation', which is important both for scientists and funding organisations and is different from their bibliographic citation. In addition to a standard literature citation, authors need a stand-alone data citation that has to be included in the publication's reference lists. Each curated dataset used in the manuscript is required to have a unique, persistent identifier, cited to credit original data generators. For this manuscript, I'd strongly recommend not using a landing page which can easily become obsolete in the near future (with PMIP5 and the beginning of CMIP7 in 2020, etc.), but creating instead a permanent site and pointing to it in the Data Availability section.

All the forcing files will be available on the PMIP4 website (https://pmip4.lsce.ipsl.fr/doku.php/exp_design:index) and the results of the simulations should be uploaded to the ESGF website. We are now including this in the data availability section.

---

## Author Comment (AC5) · 8 Feb 2019

We thank the Editor for his support and recommendation to transfer the manuscript to Geoscientific Model Development (GMD). The manuscript will thus be withdrawn from Climate of the Past. A revised version, which takes into account the Reviewers' and the short comments, will be submitted to the PMIP4 special issue of GMD.